# Fast hydrogen purification through graphitic carbon nitride nanosheet membranes

Yisa Zhou[1,3], Ying Wu[1,3], Haoyu Wu[2], Jian Xue [1]✉, Li Ding[1], Rui Wang[1] & Haihui Wang [2]✉

Two-dimensional graphitic carbon nitride (g-$C_3N_4$) nanosheets are ideal candidates for membranes because of their intrinsic in-plane nanopores. However, non-selective defects formed by traditional top-down preparation and the unfavorable re-stacking hinder the application of these nanosheets in gas separation. Herein, we report lamellar g-$C_3N_4$ nanosheets as gas separation membranes with a disordered layer-stacking structure based on high quality g-$C_3N_4$ nanosheets through bottom-up synthesis. Thanks to fast and highly selective transport through the high-density sieving channels and the interlayer paths, the membranes, superior to state-of-the-art ones, exhibit high $H_2$ permeance of $1.3 \times 10^{-6}$ mol m$^{-2}$ s$^{-1}$ Pa$^{-1}$ with excellent selectivity for multiple gas mixtures. Notably, these membranes show excellent stability under harsh practice-relevant environments, such as temperature swings, wet atmosphere and long-term operation of more than 200 days. Therefore, such lamellar membranes with high quality g-$C_3N_4$ nanosheets hold great promise for gas separation applications.

Carbon dioxide ($CO_2$) concentrations in the atmosphere have rapidly increased in the last decades due to the consumption of fossil fuels, which have caused various global climate issues[1–3]. Therefore, the Paris Agreement was reached to stipulate that carbon neutrality should be achieved in the second half of the 21st century[4]. As a carbon-free energy carrier, hydrogen is now widely considered the next generation energy to reduce $CO_2$ emissions[5]. However, most produced hydrogen is mixed with larger molecules like $CO_2$, $N_2$, $CH_4$, which needs further purification before practical applications[6]. Nowadays, traditional industrial separation methods such as distillation and pressure swing adsorption are energy-intensive. As an emerging technology, membranes separation offers an efficient energy-saving alternative for $H_2$ purification[7,8]. However, the development of traditional polymer membranes is limited by the trade-off relationship between permeability and selectivity (known as Robeson's upper bounds)[9]. Recently, two-dimensional (2D) nanosheets have offered a superb building platform for membrane construction owing to their nanometer-thin thickness[10–13]. Graphitic carbon nitride (g-$C_3N_4$) nanosheets, hosting

high-density molecular-sized pores composed of tri-s-triazine units in the entire 2D plane[14,15], are considered ideal building blocks for molecular sieving membrane. Benefits to the nanometer-thin thickness and high-density nanopores, g-$C_3N_4$ nanosheet membranes should be a superior candidate for $H_2$ separation[16,17], which was already proved by theoretical investigations. For instance, Li et al.[18] found that $H_2$ had the lowest diffusion barrier to go through the g-$C_3N_4$ nanosheets by density functional theory (DFT) calculations, which resulted in a superior selectivity between $H_2$ and other larger gases. Moreover, Guo et al.[19] predicted a high $H_2$ permeance (13 mol m$^{-2}$ s$^{-1}$ Pa$^{-1}$) across a bilayer g-$C_3N_4$ nanosheets membrane by molecular dynamics (MD) simulation. Therefore, the prospect of developing g-$C_3N_4$ nanosheets membrane is highly attractive.

Nevertheless, only a few g-$C_3N_4$ nanosheet membranes have been reported for gas separation. One reason for this is the difficulty of obtaining high-quality g-$C_3N_4$ nanosheets. Similar to other 2D materials, most g-$C_3N_4$ nanosheets are usually exfoliated from bulk materials by top-down methods[20,21]. During the top-down exfoliation processes,

[1]School of Chemistry and Chemical Engineering, Guangdong Provincial Key Lab of Green Chemical Product Technology, South China University of Technology, Guangzhou 510640, China. [2]Beijing Key Laboratory of Membrane Materials and Engineering, Department of Chemical Engineering, Tsinghua University, Beijing 100084, China. [3]These authors contributed equally: Yisa Zhou, Ying Wu. ✉e-mail: xuejian@scut.edu.cn; cehhwang@tsinghua.edu.cn

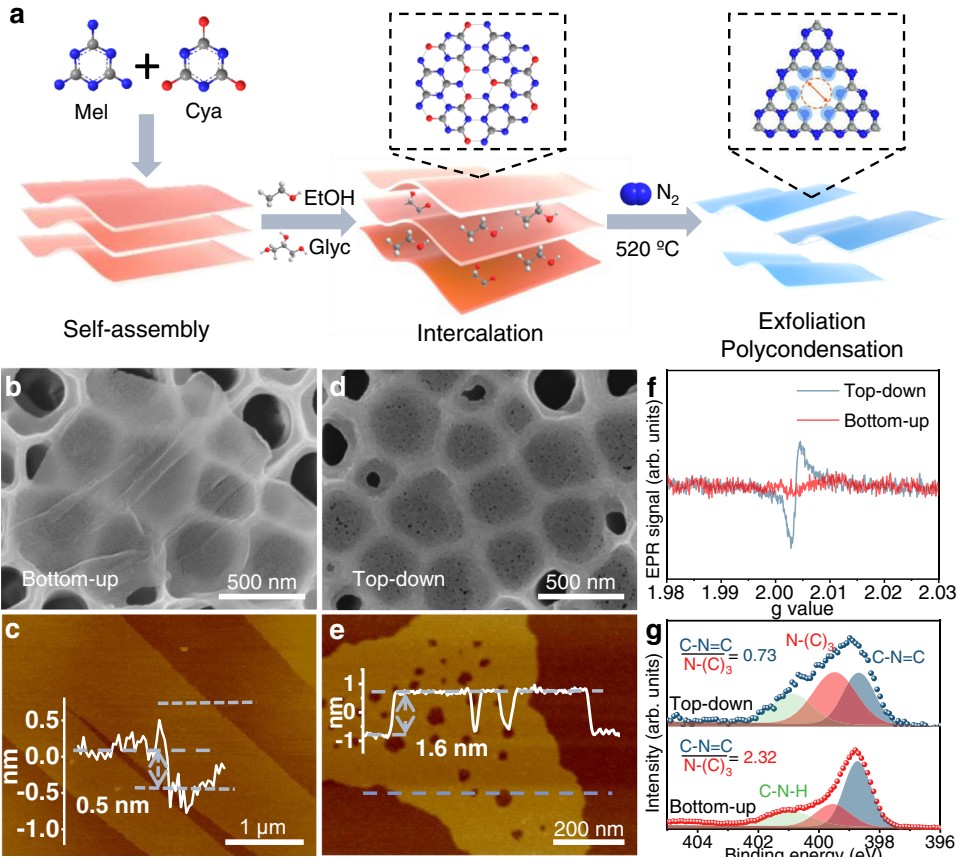

**Fig. 1 | High-quality g-C$_3$N$_4$ nanosheets. a** The fabrication process of g-C$_3$N$_4$ nanosheets. Step 1: The melamine (Mel) and cyanuric acid (Cya) were self-assembled into layered supramolecular precursors under hydrothermal conditions. Step 2: Through intercalation of an ethanol (EtOH)/glycerol (Glyc) mixture (3:1), the layers were expanded. Step 3: The g-C$_3$N$_4$ nanosheets were obtained via a calcination process under nitrogen. C, N, O, and H atoms are colored in gray, blue, red, and white, respectively. Morphology of bottom-up g-C$_3$N$_4$ nanosheets: **b** SEM image and **c** AFM image with relative height profile. Morphology of top-down g-C$_3$N$_4$ nanosheets: **d** SEM image and **e** AFM image with relative height profile. **f** Room-temperature EPR spectra and **g** N 1$s$ XPS spectra of bottom-up and top-down g-C$_3$N$_4$ nanosheets. Source data are provided as a Source data file.

harsh environments (oxidative atmosphere or acid solution) were applied to destroy the interlayer interactions in bulk g-C$_3$N$_4$, and thereby obtaining single or few-layer nanosheets[22,23]. However, the structural deterioration may occur simultaneously during this exfoliation process, leading to the formation of larger defects in g-C$_3$N$_4$ nanosheets. For example, Wang et al.[24] fabricated g-C$_3$N$_4$ nanosheet membranes containing 1.5–3 nm artificial nanopores through the top-down method using concentrated hydrochloric acid. These membranes exhibited excellent performance in nanofiltration, while the artificial nanopores were too large for H$_2$ purification. On the other hand, it was reported that g-C$_3$N$_4$ nanosheets tended to re-stack to form tight films due to strong π−π interaction[25], leading to the blockage of intrinsic in-plane nanopores. By adding polybenzimidazole chains as spaces in g-C$_3$N$_4$ nanosheets to prevent the re-stacking, Villalobos et al.[26] developed a g-C$_3$N$_4$ nanosheets-based mixed matrix membrane with considerable separation performance. However, the merits of g-C$_3$N$_4$ nanosheets can not be exploited in the mixed matrix membranes.

Hence, in this work, high-quality g-C$_3$N$_4$ nanosheets are prepared by a bottom-up method where the produced g-C$_3$N$_4$ has been delaminated into nanosheets along with the thermal polycondensation, skipping the exfoliation step, thus significantly avoiding the structural deterioration of g-C$_3$N$_4$ nanosheets. Then the isopropanol is used as a dispersant for membrane preparation to weaken the π−π interaction between g-C$_3$N$_4$ nanosheets. As a result, laminar g-C$_3$N$_4$ membranes with disordered stacking structures assembled by high-quality g-C$_3$N$_4$ nanosheets are obtained. The lamellar g-C$_3$N$_4$ nanosheet membranes

exhibit excellent H$_2$ permeance of $1.3 \times 10^{-6}$ mol m$^{-2}$ s$^{-1}$ Pa$^{-1}$ with high selectivity in mixed-gas separation. Furthermore, the gas separation mechanism through the g-C$_3$N$_4$ membrane is revealed by DFT and MD simulations, which is based on the synergistic influence of size exclusion and the interactions between gas molecules and g-C$_3$N$_4$ nanosheets. Therefore, this work not only prepares a lamellar g-C$_3$N$_4$ nanosheets membrane for efficiently H$_2$ purification but also provides a strategy for constructing other 2D nanosheets membranes for gas separation.

## Results
### Fabrication of high-quality g-C$_3$N$_4$ nanosheets
The detailed steps of the bottom-up method are shown in Fig. 1a. First, melamine (Mel) and cyanuric acid (Cya) are self-assembled into layered supramolecular precursors (Supplementary Fig. 1). Then, the ethanol (EtOH)/glycerol (Glyc) mixture are intercalated into the layered precursors. During subsequent calcination in an inert gas, the g-C$_3$N$_4$ is formed through thermal polycondensation. At the same time, the released gases (such as EtOH, Glyc, NO$_x$, and H$_2$O) produced by evaporation and decomposition of the intercalated molecules lead to the exfoliation of the layered precursors. Different from the top-down method (Supplementary Fig. 2), no further exfoliation is required, hence the structural deterioration of the g-C$_3$N$_4$ nanosheets can be greatly avoided. The obtained g-C$_3$N$_4$ nanosheets by the bottom-up method were first examined using $^{13}$C solid-state nuclear magnetic resonance (NMR). The spectrum of the g-C$_3$N$_4$ nanosheets exhibits two signal groups at 163 and 155 ppm, corresponding to the CN$_2$(NH$_x$) carbon atoms (C1) and CN$_3$ carbon atoms (C2),

respectively[27], indicating the successful synthesis of g-$C_3N_4$ (Supplementary Fig. 3). The Fourier transform infrared spectroscopy (FTIR) of both types of nanosheets (Supplementary Fig. 4a) shows the typical molecular structure of g-$C_3N_4$[14]. The signal located at 810 cm$^{-1}$ is originated from the characteristic tri-s-triazine breathing mode. The wide bands at 1690–1150 cm$^{-1}$ and 3680–2970 cm$^{-1}$ can be assigned to the stretching vibrations of C−N heterocycles and the N−H stretching vibrations of the terminal -$NH_2$/NH of g-$C_3N_4$, respectively. In the X-ray diffraction (XRD) patterns, the (100) peak at 12.9° stems from the lattice planes along c-axis due to the 2D planer disorder[28] (Supplementary Figs. 4b and 5). It was reported that the disordered g-$C_3N_4$ structure was formed by the incompletely condensed tri-s-triazine units connected by hydrogen bonds through -$NH_2$/NH at their edges[29]. However, this is not detected in the g-$C_3N_4$ nanosheets prepared using the top-down method, indicating that the in-planar atomic structure is destroyed. The differences between the two nanosheets were confirmed by scanning electron microscopy (SEM) and atomic force microscopy (AFM) images. As shown in Fig. 1b, c, g-$C_3N_4$ nanosheets with a thickness of approximately 0.5 nm are almost transparent to electron beams. In the case of the 0.326 nm g-$C_3N_4$ single layer[30], the obtained g-$C_3N_4$ nanosheet with a thickness less than double layers can be regarded as a single-layer nanostructure. The increased thickness is likely due to a "dead layer" caused by the adsorbed water between the sample and the substrate or the presence of surface adsorbates such as water molecules[31]. More importantly, no apparent nanopore defects are observed in the g-$C_3N_4$ nanosheets fabricated via this bottom-up technology (Fig. 1b, c and Supplementary Fig. 6), which is desirable for molecular sieving of gases. In contrast, large defects are inevitable in nanosheets prepared by traditional top-down thermal oxidation processes due to structural degradation. As shown in Fig. 1d, e, the g-$C_3N_4$ nanosheets obtained by the classical top-down technique contain distinct randomly distributed artificial defect pores (10–100 nm).

Furthermore, the Electron paramagnetic resonance (EPR), X-ray photoelectron spectroscopy (XPS), and element analysis (EA) of the two types of g-$C_3N_4$ nanosheets were investigated to identify defect concentrations in nanosheets. As presented in Fig. 1f, a lower EPR peak intensity in bottom-up nanosheets is observed, which can be assignable to unpaired electrons, indicating fewer defects in bottom-up g-$C_3N_4$ nanosheets[32]. The N 1s XPS spectra (Fig. 1g) shows three peaks at 398.7, 399.6, and 400.9 eV, which can be attributed to C−N=C, N−$(C)_3$, and C−NH, respectively[33]. The C−N=C to N−$(C)_3$ ratio drastically decreases from 2.32 (bottom-up method) to 0.73 (top-down approach), which also suggests a lower defect concentration in the bottom-up g-$C_3N_4$ nanosheets[34]. In the EA results, the C/N atom ratio decreases from 0.68 (bottom-up method) to 0.65 (top-down approach), further confirming fewer defects in the bottom-up nanosheets (Supplementary Table 1)[32]. In addition, all g-$C_3N_4$ nanosheets obtained via the bottom-up and top-down methods are homogenously dispersed with apparent Tyndall effects (Supplementary Fig. 7). The g-$C_3N_4$ nanosheet suspensions show a clear linear relationship between UV/Vis spectroscopic absorbance and nanosheet concentration, which can be applied to determine the nanosheet concentration for membrane fabrication (Supplementary Fig. 8).

## Fabrication of lamellar g-$C_3N_4$ membranes

It has been reported that the guest solvent can tune the stacking modes of nanosheets according to host-guest noncovalent interactions, such as steric hindrance or van der Waals interactions[35]. Therefore, in this work, using isopropanol as a guest molecule to weaken the π-π interaction between g-$C_3N_4$ nanosheets to prevent undesired restacking, the lamellar g-$C_3N_4$ membranes were prepared on porous anodic aluminum oxide (AAO) substrates (Supplementary Fig. 9). All the prepared g-$C_3N_4$ membranes are intact with no detectable pinholes or cracks and have homogeneous elemental distributions, as seen in

the SEM images (Fig. 2a, b and Supplementary Figs. 10–12). The cross-sectional transmission electron microscopy (TEM) image of the g-$C_3N_4$ membrane (Fig. 2c) reveals the turbostratic arrangement of the bottom-up nanosheets, which can be attributed to the stronger repulsive interactions weakening the π-π interactions between adjacent nanosheets (Supplementary Fig. 13). Such structures have also been reported in GO membranes[36]. While it is shown in Fig. 2d that the top-down g-$C_3N_4$ nanosheets form a compact membrane structure. Besides, considering that the aligned and unaligned stacking between adjacent layers will affect the separation applications of the membrane[37], the stacking modes of g-$C_3N_4$ nanosheets were calculated by DFT. As shown in Fig. 2e, the calculated total energy of bilayer bottom-up g-$C_3N_4$ shows that its AA stacking has minimum energy configuration, indicating that the bottom-up g-$C_3N_4$ nanosheets tend to the aligned AA stacking in the g-$C_3N_4$ membrane, forming conceivable gas-permeable interlayer pathways (Fig. 2f). In contrast, the top-down g-$C_3N_4$ nanosheets tend to be nonaligned AB stacking (Supplementary Fig. 14), which greatly reduces the effective sieving channel of g-$C_3N_4$ membrane and thus blocks the transmission of gas. Moreover, the corrugated surfaces of the g-$C_3N_4$ membrane (Supplementary Fig. 15) suggest disordered stacked nanosheets, which is consistent with the TEM results. From the 1D and corresponding 2D XRD patterns of the g-$C_3N_4$ nanosheets and membrane (Fig. 2g), the (002) diffraction peak of the g-$C_3N_4$ membrane becomes broader than that of the nanosheets, further indicating the disordered stacking of the bottom-up nanosheets in the g-$C_3N_4$ membrane[38]. Grazing incidence angle X-ray diffraction could provide more accurate XRD information of the membrane by eliminating the interference of the substrates, which also shows a broad (002) diffraction peak with a disordered turbostratic arrangement in the g-$C_3N_4$ membrane (Supplementary Fig. 16). Indeed, such lamellar disordered structure may provide additional gas transport pathways and plays a dominant role in constructing ultra-permeable membranes. For example, Yang et al.[39] reported that the performance of 2D MOF nanosheet membranes was correlated with the stacking order of the nanosheets, where the membrane with disordered stacking showed increased permeance and selectivity by 255% and 449% compared with that of ordered restacking, respectively. The XPS spectra of the g-$C_3N_4$ membrane confirms the preservation of the g-$C_3N_4$ structure in the membrane (Supplementary Fig. 17), where the characteristic peak of g-$C_3N_4$ membrane is similar to that of the prepared nanosheets. In addition, the g-$C_3N_4$ membrane exhibits the Young's modulus up to 159 MPa, revealing its excellent mechanical property (Supplementary Fig. 18 and Supplementary Table 2).

## Gas separation performance of g-$C_3N_4$ membranes

The gas separation performance of the g-$C_3N_4$ membranes were measured systematically using Wicke-Kallenbach permeation cells (Supplementary Fig. 19). The $H_2$ flux through a 1 μm-thick g-$C_3N_4$ membrane reaches high permeance of 3.3 × 10$^{-7}$ mol m$^{-2}$ s$^{-1}$ Pa$^{-1}$ (Fig. 3a, b and Supplementary Fig. 20). As the kinetic diameter of the molecules increases, the gas permeance decreases sharply, indicating a clear cut-off between $H_2$ and the other tested gases. The $H_2$ to $CO_2$, $N_2$, $CH_4$, $C_3H_6$, and $C_3H_8$ selectivity are 41, 23, 21, 83, and 113, respectively, which far exceeds the Knudsen selectivity (Supplementary Table 3). However, the 1-μm-thick g-$C_3N_4$ membranes assembled by the top-down g-$C_3N_4$ nanosheets exhibit similar selectivity to the Knudsen diffusion (Supplementary Fig. 21). Considering the same material and thickness of the two types of membranes, their performance differences are attributed to membrane defects. The g-$C_3N_4$ membrane assembled by the bottom-up g-$C_3N_4$ nanosheets exhibits noticeable separation performance while the one using the top-down nanosheets does not, indicating that the defects have a significant adverse effect on membrane performance. Recently, Li et al.[40] developed a strategy of vapor linker exchange

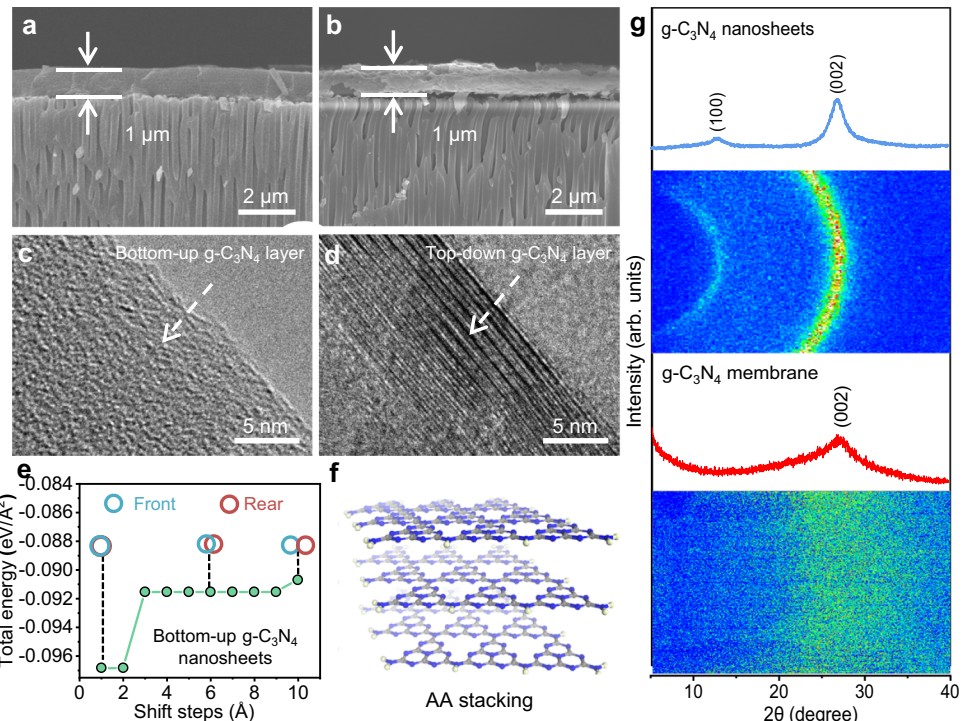

**Fig. 2 | g-C₃N₄ nanosheets membranes with disordered layer-stacking structure.** Cross-sectional SEM images of the **a** bottom-up g-C₃N₄ and **b** top-down g-C₃N₄ membranes. Cross-sectional TEM images of the **c** bottom-up g-C₃N₄ and **d** top-down g-C₃N₄ membranes. **e** The DFT calculations about stacking states of bottom-up g-C₃N₄ nanosheets in g-C₃N₄ membranes. The front represents the first layer of nanosheet in a two-layer system of g-C₃N₄ and the rear represents another layer of nanosheet below the first layer of nanosheet. **f** The AA stacking of bottom-up g-C₃N₄ nanosheets. C, N, and H atoms are colored in gray, blue, and white, respectively. **g** 1D and corresponding 2D wide-angle XRD patterns of bottom-up g-C₃N₄ nanosheets and g-C₃N₄ membrane. Source data are provided as a Source data file.

inducing partial amorphization to conglutinate grain boundary/crack defects of membranes, where the ZIF-8 composite membrane showed competitive $H_2/CO_2$ selectivity up to 2400, which also indicated the great significance of restraining membrane defects, further illustrating the advantages of the bottom-up method for the synthesis of g-C₃N₄. Gas permeation through the g-C₃N₄ membrane is different from Knudsen diffusion and is mainly governed by the kinetic gas diameter rather than its molecular weight, known as the size exclusion or molecular sieving mechanism[41] (Supplementary Fig. 22). Besides, the g-C₃N₄ membranes with different thicknesses were fabricated (Supplementary Fig. 23 and Supplementary Table 4). As the membrane thickness increases, the $H_2$ permeance of the g-C₃N₄ membrane decreases and the $H_2/CO_2$ selectivity increases, which is characteristic of molecular separation membranes. The 300-nm-thick g-C₃N₄ membranes exhibit an $H_2$ flux of $1.3 \times 10^{-6}$ mol m$^{-2}$ s$^{-1}$ Pa$^{-1}$ with an $H_2/CO_2$ separation factor of 16. The results are promising for industrial applications as enhancing permeance is more important than improving selectivity to reduce separation costs[42]. Indeed, the g-C₃N₄ membranes show ultra-high $H_2$ flux with considerable selectivity compared to the other membranes reported in the literatures (see Supplementary Table 5), which breaks the conventional upper bound, as shown in Fig. 3c. Moreover, the g-C₃N₄ membrane exhibits excellent stability during a 1000 h continuous operation. After being stored in an ambient environment without the introduction of any protective gas at room temperature for 200 days, the membrane was examined again, whose $H_2/CO_2$ separation performance was still stable and as high as the fresh one as shown in Fig. 3d. Furthermore, the gas performance of the g-C₃N₄ membranes were further investigated in harsh environments, including water vapor, elevated temperatures and pressures (Supplementary Fig. 24)[43]. When exposed to a 3 vol% water vapor environment, the membrane can operate stably for 100 h at room temperature (Supplementary Fig. 25). After three

temperature cycles between 25 and 150 °C, the separation performance of the g-C₃N₄ membrane is recovered (Supplementary Figs. 26–28), which can be attributed to the excellent thermostability of the g-C₃N₄ material[22] (Supplementary Fig. 29). Additionally, the g-C₃N₄ membrane was also evaluated by durability testing with an equimolar $H_2/CO_2$ mixture feed at 120 °C in the presence of water vapor (water activity of 0.353, Supplementary Note 1). It can be seen that the g-C₃N₄ membrane performs well for 100 h even under wet gas mixture conditions at elevated temperatures (Supplementary Fig. 30), indicating the good hydrothermal stability of the g-C₃N₄ membrane. The stability of g-C₃N₄ system can also be confirmed by MD simulation. It can be seen that g-C₃N₄ system shows very small energy fluctuations in the long-term MD simulations (Supplementary Fig. 31). This is consistent with the substantially unchanged schematic diagrams of the g-C₃N₄ layer before and after MD simulations, also reflecting its good structural stability[44] (Supplementary Fig. 32). Moreover, when the feed pressure increases to 2 bar (transmembrane pressure: 1 bar), the g-C₃N₄ membrane exhibits a decreased $H_2/C_3H_8$ separation factor of 17, which can be recovered to the initial value basically after releasing the pressure, as shown in Fig. 3e. However, the $H_2$ and $CO_2$ permeances increase rapidly while the $H_2/CO_2$ selectivity decreases with the increased feed pressure (Fig. 3f). The possible reason behind the decrease in selectivity at high pressure might be the existence of the parallel non-selective transport pathways where viscous diffusion is prevalent and which dominates the membrane performance when the transmembrane pressure difference is greater than zero[45,46].

## Discussion

The permeances of $H_2$, $CO_2$, $N_2$, and $CH_4$ do not completely follow the order of kinetic diameters of gas molecules, indicating gas transport behaviors are not only affected by molecular size. Therefore, the

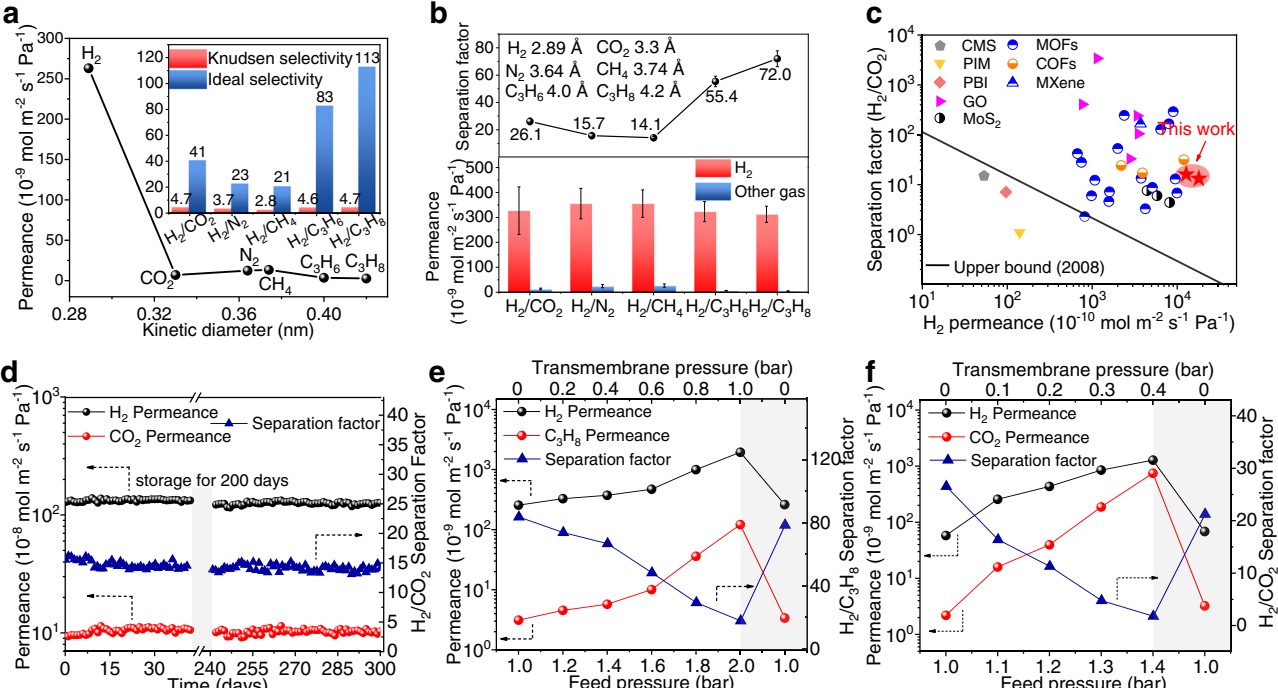

**Fig. 3 | Gas separation performance through the g-C₃N₄ membranes. a** Single gas permeation of the 1-µm-thick g-C₃N₄ membrane at room temperature and 1 bar. The inset shows the gas selectivity for H₂ over other gases. **b** Permeance and separation factors of the 1-µm-thick g-C₃N₄ membranes in the equimolar mixed-gas permeation at room temperature and 1 bar. Inset shows the kinetic diameters of various gas molecules. Errors bars indicate the standard deviation of three measurements. **c** Comparison of H₂/CO₂ separation performance of g-C₃N₄ membranes with state-of-the-art membranes at room temperature. **d** Long-term stability of the g-C₃N₄ membrane for equimolar H₂/CO₂ mixture. The gray areas represent the membrane was stored in an ambient environment without the introduction of any protective gas at room temperature for 200 days. **e** Gas permeances and H₂/C₃H₈ separation factor of the g-C₃N₄ membrane as a function of the feed pressure at room temperature. The gray areas show the performance of g-C₃N₄ membrane after releasing the pressure. **f** Gas permeances and H₂/CO₂ separation factor of the g-C₃N₄ membrane as a function of the feed pressure at room temperature. The gray areas show the performance of g-C₃N₄ membrane after releasing the pressure. Source data are provided as a Source data file.

behaviors of gas molecules passing through the g-C₃N₄ layer were investigated using DFT. Considering the universality, we still used the most typical g-C₃N₄ structure to perform the simulation process. It is found that when $H_2$, $CO_2$, $N_2$, and $CH_4$ molecules pass through the g-C₃N₄ layer, the electron clouds of gas molecules partially overlap with the atoms around the g-C₃N₄, as shown in the partial density of states (PDOS)[47] results (Supplementary Fig. 33). As a result, $H_2$, $CO_2$, $N_2$, and $CH_4$ molecules can pass through g-C₃N₄ layer after the partial overlap of electron clouds of the gas molecules and the g-C₃N₄. However, in virtue of a large number of negatively polarized N atoms in the g-C₃N₄ nanosheets (red reflects negatively polarized sites, blue represents the opposite), g-C₃N₄ nanosheets show stronger electrostatic interactions with $CO_2$ molecules with the deepest blue[48], which increases the resistance to $CO_2$ diffusion and results in a high separation factor of $H_2/CO_2$ (Fig. 4a). While $CH_4$ with light blue possesses relatively weak interactions with g-C₃N₄ nanosheets, resulting in relatively fast $CH_4$ diffusion. This is consistent with N-functionalized graphene membranes reported previously[49]. When the carbon atoms were substituted by nitrogen in the porous graphene membrane, the properties of the nanosheets changed and influenced the gas permeability. Consequently, the calculated energy barriers ($E_b$)[18] for gas molecules across the nanosheets are 0.132, 0.969, 0.782, and 0.791 eV for $H_2$, $CO_2$, $N_2$, and $CH_4$, respectively (Fig. 4b and Supplementary Table 6), which suggests that g-C₃N₄ is promising for sieving $H_2$ from these larger molecules, especially $CO_2$. The adsorption isotherms of $H_2$, $CO_2$, $N_2$, and $CH_4$ on the g-C₃N₄ membranes at room temperature also indicate that the g-C₃N₄ membranes tend to preferentially adsorb $CO_2$ compared to other gases, such as $H_2$, $N_2$, $CH_4$ (Supplementary Fig. 34), delaying the $CO_2$ transport[50]. However, the permeation of $CH_4$ is slightly higher than that of $N_2$ in the experiment, which is different from the simulation results. This may be

because the deviation in the experimental measurement or $CH_4$ as a spherical molecule[51] is easier to pass through. It should be noted that the g-C₃N₄ membranes exhibit a sharp separation of $H_2/C_3$ with higher gas selectivity for $H_2/C_3H_6$ and $H_2/C_3H_8$ (exceeding 80), which can be attributed to the size sieving effect due to the larger size of $C_3H_6$ (4.0 Å) and $C_3H_8$ (4.2 Å) molecules[52]. These results show that the gas separation mechanism of the g-C₃N₄ membrane is based on the synergistic effect of size exclusion and the interactions between gas molecules and g-C₃N₄ nanosheets.

Moreover, MD simulations were conducted to study the gas transport through the g-C₃N₄ membrane. In $H_2/CO_2$ mixed gas simulations (Fig. 4c), it is found that all $H_2$ molecules have moved quickly through the g-C₃N₄ layers at the very beginning, yielding a diffusivity ratio of 130:4 for $H_2$:$CO_2$ with the $H_2/CO_2$ selectivity of 32.5 at 1.0 ns. The predicted selectivity decreases with extended time because there is no $H_2$ molecule on the feed side while $CO_2$ continues to pass through in the subsequent simulation. Besides, the selectivity predicted by the DFT and MD simulations shows several orders of magnitude differences (see Supplementary Note 2), which may be that DFT calculations use different reference states for the energies (separate nucleus, electrons) while potential energy with classical MD simulation only includes the intra and intermolecular potential energies and not the piece due to internal part of the molecular partition function. There is a great difference between the two calculation methods. Therefore, in this work, there is not a strict one-to-one correspondence between the results by DFT calculations and MD simulations, as reported by Wang et al.[53]. Even so, the MD results qualitatively agree well with the DFT results. With respect to $H_2/C_3H_6$ mixtures (Fig. 4d), $H_2$ quickly passes through the g-C₃N₄ layer while $C_3H_6$ does not, exhibiting a sharpening sieving effect for $H_2/C_3H_6$.

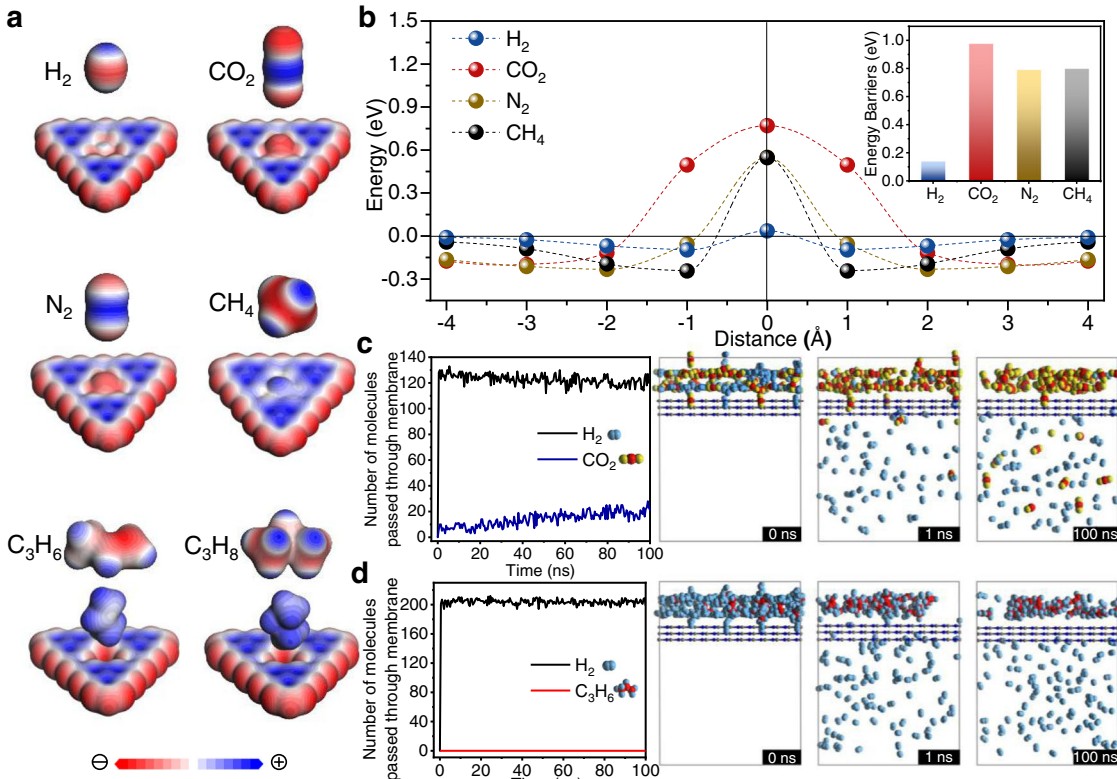

**Fig. 4 | Gas separation mechanism of the g-C₃N₄ layer. a** Schematic illustrations of gas molecules with electrostatic potentials distribution through the g-C₃N₄ layer. **b** Potential energy surfaces for $H_2$, $CO_2$, $N_2$, and $CH_4$ on the g-C₃N₄ layer. Inset is the energy barrier for gas molecules across the monolayer g-C₃N₄. The number of gas molecules of **c** $H_2$/$CO_2$ mixtures and **d** $H_2$/$C_3H_6$ mixtures through the g-C₃N₄ layer in molecular dynamics (MD) simulations as a function of simulation time. Simulation snapshots at 0, 1, and 100 ns for $H_2$/$CO_2$ mixtures and $H_2$/$C_3H_6$ mixtures are given. C, O, and H atoms are colored in red, yellow, and blue, respectively. Source data are provided as a Source data file.

In this work, lamellar g-C₃N₄ nanosheet membranes with excellent gas separation performances are successfully constructed. The membranes are assembled using high-quality nanosheets obtained through a bottom-up method, allowing the exfoliation step to be skipped, thus maintaining structural integrity. Guest molecules are introduced during membrane preparation to prevent re-stacking of the g-C₃N₄ nanosheets. The fabricated laminar g-C₃N₄ nanosheet membranes exhibit $H_2$ permeance of up to $1.3 \times 10^{-6}$ mol m⁻² s⁻¹ Pa⁻¹ due to the synergistic effect of the high-density sieving channels and disordered stacking structure. In addition, the gas separation mechanism of the g-C₃N₄ membrane is based on the integrated gating effects of size exclusion and the interactions between gas molecules and g-C₃N₄ nanosheets, as evidenced by both the experimental results and computational simulations. These membranes also show good stability even under harsh practice-relevant environments. The excellent separation performance with long-term stability enables the g-C₃N₄ membrane to serve as a promising candidate for $H_2$ purification, offering an opportunity for the development of carbon neutrality.

## Methods
### Materials
Melamine (99%, Aladdin), phosphoric acid (≥85%, Guangzhou Chemical Reagent Factory), isopropanol (Guangzhou Chemical Reagent Factory), ethanol (Runjie Chemistry), glycerol (Runjie Chemistry), AAO substrate with a diameter of 15 mm (with a pore size of 160–200 nm, PuYuan Nanotechnology Limited Company). All of the materials were used as purchased without further purification.

### Preparation of the g-C₃N₄ nanosheets
First, a bottom-up method was employed to fabricate the g-C₃N₄ nanosheets[25], as follows. Melamine (1 g) and phosphoric acid (1.2 g) were dissolved in 100 mL of deionized water at 80 °C in a thermostatic water bath until the melamine dissolved completely. Then the solution was transferred into a hydrothermal reactor and heated at 180 °C for 10 h. After centrifugation at 2795 × g for 30 min, the obtained mixture was dried in an oven at 60 °C. The layered precursors were obtained and then refluxed for 3 hours at 90 °C with a 20 mL mixed solution of ethanol and glycerol (3:1, v-v). The powders were washed with ethanol several times, and then the layered precursors after intercalation were obtained after dried at 60 °C. Here, to keep the inherent nanoporous structures of g-C₃N₄, we chose $N_2$ to achieve an inert operation condition. It has been reported that the gas atmosphere affects the nanoporous structures of g-C₃N₄, and more complete g-C₃N₄ nanoporous structures were obtained in an inert atmosphere.[54] Aminabhavi et al.[55] found that in the process of synthesizing g-C₃N₄, ammonia was obtained in melamine decomposition and made g-C₃N₄ produce N-defects. So the layered precursors after intercalation were heated to 520 °C under $N_2$ condition with the heating rate of 2 °C min⁻¹ and kept for 2 h. Then high-quality g-C₃N₄ nanosheets of 6–8 mg can be obtained. For comparison, a traditional top-down method was also employed to fabricate the g-C₃N₄ nanosheets.[56] 10 g melamine was heated to 520 °C for 6.5 h under an air atmosphere with a heating and cooling speed of 5 °C min⁻¹. Then the obtained g-C₃N₄ powders (1 mg) were dispersed in the water (50 mL) to obtain the g-C₃N₄ suspension at the concentration of 0.02 mg mL⁻¹. The g-C₃N₄ nanosheets suspension was obtained by ultrasonic treatment for 2 h.

## Preparation of the g-C$_3$N$_4$ membranes

For assembling nanosheets into membranes by vacuum filtration, the g-C$_3$N$_4$ nanosheets powders (1 mg) prepared by the bottom-up method need to be dispersed in isopropanol (50 mL) to obtain the nanosheets suspension. The nanosheets suspension was treated by ultrasonication for 5 min for better dispersion. Then the g-C$_3$N$_4$ membranes were fabricated on AAO substrates by filtering a certain amount of two different g-C$_3$N$_4$ nanosheets suspensions, respectively. Next, these g-C$_3$N$_4$ membranes were placed in a vacuum drying chamber at room temperature for more than 12 h to remove residual solvents in the membranes.

## Gas permeation test

All the gas permeation tests were performed in a homemade Wicke-Kallenbach apparatus[41]. The gases with different kinetic diameters were used as the feed gas, while Ar was used as the sweep gas. The gas volumetric flow rate was constantly controlled at 50 mL min$^{-1}$ for single and mixed gas tests. Gas chromatography (GC Agilent 7890) was employed to obtain the gas concentrations of permeate gas. The gas flow was controlled using mass flow controllers (MFCs) and corrected by a bubble flowmeter. The membrane module was packed with heating tape, and thermocouple and temperature controller devices were used to control the temperature and heating rate (2 °C min$^{-1}$). Feed gas was saturated with water vapor before feeding to the membrane module. The partial pressure of water vapor in the feed gas was varied by controlling the temperature of the water tank. The permeate stream was chilled in an iced cold trap. The relative humidity (RH) of the feed stream was measured by a humidity transmitter.

At equilibrium conditions, the water activity ($a_w$) is calculated by Eq. (1):

$$a_w = \frac{RH}{100} = \frac{P_{H_2O}}{P_{sat}} \tag{1}$$

where $P_{H_2O}$ is the water vapor partial pressure, $P_{sat}$ is the saturation water vapor pressure at the stream temperature and pressure[57].

The following equations calculate the gas permeance ($P_i$, mol m$^{-2}$ s$^{-1}$ Pa$^{-1}$) and ideal selectivity $S_{i/j}$,

$$P_i = \frac{N_i}{\Delta P_i \cdot A} \tag{2}$$

$$S_{i/j} = \frac{P_i}{P_j} \tag{3}$$

where $N_i$ (mol s$^{-1}$) is the permeate flow rate of the component gas i, $\Delta P_i$ (Pa) is the transmembrane pressure difference of i, and $A$ (m$^2$) is the membrane area.

The mixed gas separation factor ($\alpha_{i/j}$) is calculated as follows:

$$\alpha_{i/j} = \frac{y_i/y_j}{x_i/x_j} \tag{4}$$

where $x_i$ and $x_j$ are the volumetric fractions of component i and component j at the feed side, respectively; $y_i$ and $y_j$ are corresponding volumetric fractions at the permeate side.

## DFT simulations

All the computational simulations were performed using the Materials Studio 7.0 package[58]. To determine the stable stacking status mode of nanosheets in g-C$_3$N$_4$ membranes, we constructed a two-layer system of g-C$_3$N$_4$ corresponding to both of them. One layer shifted to specific extents related to the other layer. The total energy of these two-layer systems is calculated in the Forcite module. The van der Waals (vdW) interactions between layers are described by Dreiding force field[59],

with cubic spline and a cutoff distance of 12.0 Å for vdW truncation. The electrostatic interactions are modeled by Ewald[60] for the summation of atomic charges of g-C$_3$N$_4$ calculated using the QEq method, with the accuracy set to be $1 \times 10^{-4}$ kcal/mol. The framework of g-C$_3$N$_4$ is assumed to be rigid in the calculations.

The PDOS analysis of g-C$_3$N$_4$ and gas molecules (H$_2$, CO$_2$, N$_2$, and CH$_4$) were computed by GGA/PBE level of functional under DNP basis set in Dmol$^3$ module[61], where DFT-D correction was considered with Grimme method[62], and the core electrons were treated by an all-electron method. During the simulations, $1.0 \times 10^{-6}$ Ha of self-consistent field (SCF) tolerance and 500 SCF cycles were used, and 0.05 Ha of thermal smearing were applied to orbital occupation to speed up convergence. The periodic boundary conditions were considered in DFT calculations.

With the same simulation level of PDOS, the electrostatic potential distributions of gas-free g-C$_3$N$_4$ and gas molecules (H$_2$, CO$_2$, N$_2$, CH$_4$, C$_3$H$_6$, and C$_3$H$_8$) were computed. In addition, the potential energy surfaces were obtained by calculating the interaction energies between gas molecules and g-C$_3$N$_4$ nanosheets at different distances from the nanosheets. Typically, a gas molecule was placed at several positions on the nanosheets, and the mass center of the gas was fixed in the z-direction (perpendicularly to the g-C$_3$N$_4$ layer). The interaction energy ($E_{int}$) of the gas molecules with g-C$_3$N$_4$ at the corresponding position is calculated by Eq. (5).

$$E_{int} = E_{g-C_3N_4 + gas} - \left( E_{g-C_3N_4} + E_{gas} \right) \tag{5}$$

where $E_{g-C_3N_4 + gas}$ is the total energy of the g-C$_3$N$_4$-gas configuration, and $E_{g-C_3N_4}$ and $E_{gas}$ are the single point energy of g-C$_3$N$_4$ and the gas molecules, respectively.

Next, the $E_b$ (energy barrier) is defined as the difference between the interaction energies at the transition state (TS, z = 0) and the stable state (SS, the stable adsorption state when the attractive interaction is maximum and z ≠ 0) of the gas molecule on g-C$_3$N$_4$[26].

$$E_b = E_{TS} - E_{SS} \tag{6}$$

where $E_{TS}$ and $E_{SS}$, respectively, stand for the TS energy and SS energy when a gas molecule permeates through the g-C$_3$N$_4$ nanosheet. This modeling method has been successfully used to explore the interaction mechanism between porous materials and small gas molecules[26].

## MD simulations

All the computational simulations were performed using the Materials Studio 7.0 package[58]. The gas permeation of H$_2$/CO$_2$ and H$_2$/C$_3$H$_6$ mixtures through the g-C$_3$N$_4$ layer was modeled by MD simulations. For H$_2$/CO$_2$ mixtures, 260 gas molecules (130 for each gas species) were placed in the feed chamber in the simulation. For H$_2$/C$_3$H$_6$ mixtures, 420 gas molecules (210 for each gas species) were placed in the feed chamber. Then gas mixtures were loaded into the two-layer system of g-C$_3$N$_4$ employing the Sorption module with the Metropolis method, where $1 \times 10^6$ kPa of pressure is fixed for each gas. The Dreiding force field and interaction settings were the same as the energy calculations of the gas-free g-C$_3$N$_4$ system. The loaded gas molecules were then optimized with a Forcite module for energy minimization of the adsorbent-adsorbate system, while the atom of g-C$_3$N$_4$ was kept rigid. Subsequently, MD simulations were performed with NVT ensemble in the Forcite module at 298 K, where $1 \times 10^5$ ps of total simulation time was used with a time step of 1.0 fs. The Berendsen method[63] was applied to maintain the temperature. During the MD simulation, the Dreiding force field and Ewald summation describe the vdW and electrostatic interaction between the gas and g-C$_3$N$_4$. The periodic boundary conditions were considered in three dimensions. MD simulations were run for 10,000 ps to study the potential energy

fluctuations of the g-C$_3$N$_4$ system with 298 K temperature to verify the stability of the g-C$_3$N$_4$ system.

## Characterization

NMR measurements were recorded on an AVANCE III spectrometer (Bruker) operating at a proton frequency of 400 MHz. XRD patterns were recorded under ambient conditions with a Bruker D8 Advance diffractometer with Cu Kα radiation at 40 kV and 40 mA. The two-dimensional XRD patterns of the g-C$_3$N$_4$ membrane and g-C$_3$N$_4$ powder were obtained by XRD (Rigaku Smart Lab X-Ray Diffractometer) equipped with a 2D X-ray detector using Cu Kα radiation source from 5° to 40° under two-dimensional detection mode. The microstructure of the membranes was observed by the SEM using a HITACHI SU8200. The AFM images were obtained using a Bruker MultiMode 8 scanning probe microscope (SPM, VEECO) in the tapping mode. The room-temperature EPR spectra were measured with Bruker A300. The XPS analysis was performed using an ESCALAB 250 spectrometer (Thermo Fisher Scientific) with monochromated Al Kα radiation (1486.6 eV) under a $2 \times 10^{-9}$ Torr pressure. The concentrations of the g-C$_3$N$_4$ nanosheets dispersions were measured by UV-vis spectrum (Shimadzu UV-2450). TEM images were obtained using a JEOL JEM-2100F microscope with an acceleration voltage of 200 kV. FTIR (Nicolet 5700 spectrometer) was used to detect the characteristic stretching vibration modes of the sample. Elemental analysis was measured on a Vario EL cube elementary. Thermogravimetric (TG) measurement was analyzed on a Netzsch STA 449F3 instrument under the flow of N$_2$. The adsorption isotherms of H$_2$, CO$_2$, N$_2$, and CH$_4$ on the g-C$_3$N$_4$ membranes were measured using a Micromeritics (ASAP 2460) instrument. The mechanical properties were measured by using an Instron-5565 universal tensile testing machine (USA).

## Data availability

Further data that support the findings of this study are available on request from the corresponding author. Source data are provided with this paper.

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

## Acknowledgements

J.X. acknowledges the funding from the National Key R&D Program of China (Grant No. 2020YFB1505603), the Natural Science Foundation of China (22075086), and the Guangdong Basic and Applied Basic Research Foundation (2022A1515010980). H.H.W. acknowledges the funding from the Natural Science Foundation of China (22138005, 22141001).

## Author contributions

Y.Z., J.X., and H.H.W. conceived the idea and designed the experiments. Y.Z. finished the experiments and performed the characterization of membrane performance with help of J.X., R.W., and L.D. Y.W. contributed to the computational simulations. Y.Z., H.Y.W., J.X., and H.H.W. wrote the manuscript.

## Competing interests

The authors declare no competing interests.
