## [Peer Review File · Nature Communications]

Fast hydrogen purification through graphitic carbon nitride nanosheet membranesReviewer #1 (Remarks to the Author):

In the manuscript entitled "Ultra-fast hydrogen purification through carbon nitride nanosheet membranes", it is shown that lamellar C₃N₄ nanosheets can be used as gas separation membranes with a disordered layer-stacking structure. Authors revealed that these membranes show outstanding stability under harsh practice-relevant environments, such as temperature swings, wet atmosphere and long-term operation. Work is well-written and all the required details on the process are presented, for example, the preparation of the C₃N₄ membranes is described as well as DFT and MD simulation methodology. The results are of high interest for the understanding of gas separation methods. In my opinion, it can be accepted in the present form.

Reviewer #2 (Remarks to the Author):

This manuscript reports high-aspect ratio and thin g-C₃N₄ nanosheets by a bottom-up approach. The nanosheets are used to form thin membranes and attractive permselective H₂ separation is demonstrated. The strength of the manuscript lies in developing and demonstrating high quality membranes from g-C₃N₄ nanosheets. Rigorous tests are done, and membrane stability is demonstrated expect that the feed pressure is not clearly mentioned (see comment below). However, in its current form, the manuscript has several inconsistencies including several conclusions which do not seem right.

The manuscript describes that 3.1 Å holes in the tris-triazine ring are responsible for H₂ permeation. However, no evidence of the presence of such structure is given. The reported material is amorphous based on the X-ray diffraction data. It is not clear as to why one can consider an exact structure (tris-triazine) shown in Fig. 4A to predict transport from the membrane. Author do carry out FTIR and NMR to understand the structure. In NMR, based on the tris-triazine structure, the peak height of C1 and C2 carbon should be equal; however, the data suggests something very different. In FTIR, author only discuss a single triazine ring.

Assuming tris-triazine nanopores are responsible for transport, author carried out DFT simulation of molecules crossing through the nanopore. The DFT calculation predicts a much higher activation energy of CO₂ (0.986 eV) compared to methane (0.694). This is unexpected because CO₂ is a much smaller molecule than methane. Also, based on MD simulation, approximately 120 H₂ and 20 CO₂ molecules cross the pore in 100 ns simulation. This is surprising because the energy barrier for CO₂ is close to 1 eV and in fact with such time scale and high energy barrier, no CO₂ should cross over.

MD simulation predicts a H₂/CO₂ selectivity of 6 which is not discussed. This is several orders of magnitude smaller than the selectivity one expects from DFT data (i.e. ratio of $\exp(-0.397 \text{ eV}/kT)$ for H₂ and $\exp(-0.986 \text{ eV}/kT)$ for CO₂). No explanation is provided. I could not find the description of feed pressure for the membrane study. If feed is not pressurized, the data should be reported from at least mildly pressurized feed (transmembrane pressure difference of 2 bar). This is because literature which indicates that the performance of laminar membranes can be sensitive to the feed pressurization.

Overall describing the structure of amorphous g-C₃N₄ is extremely challenging. I am of the opinion that it is perhaps not necessary to assume a certain structure because it confuses the field. I therefore recommend (i) either providing more direct proof of structure, or (ii) removing description of structure as made of 3.1 Å pores.

Minor comments

"However, the development of traditional polymer or zeolite membranes is limited by the trade-off relationship between permeability and selectivity (known as Robeson's upper bounds)". Please correct this as Robeson upper bound is only for polymers.

The Fourier transform infrared spectroscopy (FTIR) of both types of nanosheets (Supplementary Fig. 4a) showed signals corresponding to the in-plane organization of tri-s-triazine units in C₃N₄. I do not think that FTIR show in-plane organization but rather FTIR signal from tri-s triazine units.

“the peak at 12.9° corresponding to the (100) periodic repetition of tri-s-triazine verifies the existence of triangular nanopores in the C₃N₄”; This should be explained in terms of structure or schematic. The material prepared is an amorphous material. It is not clear as to what structure relates to the (100) peak. Further, it is not clear as to how this peak relates to pore or pore shape as there is lack of in-plane order in the synthesized material.

Reviewer #3 (Remarks to the Author):

Dear editor

Zhou et al performed an interesting experimental-theoretical study on H₂ permeability and gas selectivity on carbon nitride nanosheets. The study is comprehensible and well-written, it deserves publication in Nature communication after the authors give full attention to the following comments and suggestions:

Major revision

1. On page 6, the authors mentioned results coming from their DFT calculations. Nevertheless, such information is abruptly given. It is then recommended that the DFT output mentioned in this section should be adequately presented prior to be invoked and compared with experimental data. Such results (line 150-151), as they are given in the manuscript are not understood, and they are challenging to interpret.

2. If the Top-down and Bottom-up methods were somehow simulated with the theoretical methodologies, the corresponding molecular arrangements and discussion of the results should be presented in the main text of the manuscript. It is highly recommended that Fig. S14 and S15 be presented in the main text.

3. For completeness of the study, it is suggested that the authors perform MD simulations to test the stability of the C₃N₄ system; i.e., geometry restrictions should be omitted. This could be performed only with a representative system.

4. On page 10, lines 236-238 the authors stated: “CO₂, N₂, and CH₄ molecules (slightly larger than 3.1 Å) can pass through the pores because of the partial overlap of electron clouds of the gas molecules and the atoms around the C₃N₄ triangular pore”. To give insight into this assumption, please map the PDOS of C₃N₄/gas molecules systems to quantify the degree of overlapping.

5. On page 10, lines 239-240: There seems to be an anomaly with Fig. 4(c), since the E_b of the reaction pathways do not correspond to the E_b of the column graph shown in the inset. If the curves correspond to reaction pathway calculations, please give the Computational details of the method used.

6. On page 10, lines 243-244, the authors stated: “Obviously, their transport behaviors are not only affected by molecular size but also by interactions between the gas molecules and C₃N₄ nanosheets”.

This is not necessarily obvious; the authors are requested to verify how the electronic states are overlapped among the C₃N₄ and the different gas molecules. Additionally, the authors should comment Figs. S31 and S32 and deepen into the issue. It is not enough to only indicate where the figure is.

7. Regarding the DFT computational details (page 14): Although the methodology is correctly applied, this reviewer suggests that the authors consider the border effects; i.e., the truncation of C₃N₄ and the addition of H-atoms to complete valences may play a non-beneficial role in the computations. The authors are urged to perform a calculation in which periodic boundary conditions are included, where the border effects are excluded.

8. On page 15, line 366: Please give more specifications of the force field. Is it universal? Which module in Materials Studio was implemented to perform the MD simulations?

Reviewer #4 (Remarks to the Author):

In this manuscripts, the authors reported on the C₃N₄-based membrane for selective H₂ filtration against several other gas molecules. The work is overall of high importance, using C₃N₄ as the active layer in the filtering membrane which shows high performance in gas separation and purification. However, I have the below concerns before considering the manuscript to be published:

- 1) The C₃N₄ is claimed to be "defect-free", which is quite ambiguous and unfair since C₃N₄ has certain intrinsic structural defects arising from high temperature synthesis. No strong evidence is given to show the material is "perfect"**
- 2) Any adsorption of H₂ or other gases by the membrane?**
- 3) How about the mechanical strength of the membranes?**
- 4) Can the membranes work at elevated temperatures?**

Response to the Reviewers' Comments

Many thanks to the reviewers for their valuable comments and suggestions. The followings are the point-by-point answers to the concerns:

Response to Reviewer 1

In the manuscript entitled "Ultra-fast hydrogen purification through carbon nitride nanosheet membranes", it is shown that lamellar C_3N_4 nanosheets can be used as gas separation membranes with a disordered layer-stacking structure. Authors revealed that these membranes show outstanding stability under harsh practice-relevant environments, such as temperature swings, wet atmosphere and long-term operation. Work is well-written and all the required details on the process are presented, for example, the preparation of the C_3N_4 membranes is described as well as DFT and MD simulation methodology. The results are of high interest for the understanding of gas separation methods. In my opinion, it can be accepted in the present form.

Response: We are greatly thankful to the reviewer for the encouragement on our work and his/her valuable suggestions.

Response to Reviewer 2

This manuscript reports high-aspect ratio and thin g- C_3N_4 nanosheets by a bottom-up approach. The nanosheets are used to form thin membranes and attractive permselective H_2 separation is demonstrated. The strength of the manuscript lies in developing and demonstrating high quality membranes from g- C_3N_4 nanosheets. Rigorous tests are done, and membrane stability is demonstrated expect that the feed pressure is not clearly mentioned (see comment below). However, in its current form, the manuscript has several inconsistencies including several conclusions which do not seem right.

Response: Thanks for your comments. According to your comment, we have supplemented the research on the effect of trans-membrane pressure on the separation performance of our C_3N_4 membrane and corrected the wrong analysis of the C_3N_4 structure and the XRD and FTIR results of C_3N_4 in the revised manuscript. We have revised our manuscript prudently according to your suggestions, and the detailed point-to-point responses to the comments are shown below.

1. The manuscript describes that 3.1 Å holes in the tris-triazine ring are responsible for H_2 permeation. However, no evidence of the presence of such structure is given. The reported material is amorphous based on the X-ray diffraction data. It is not clear as to why one can consider an exact structure (tris-triazine) shown in Fig. 4A to predict transport from the membrane. Author do carry out FTIR and NMR to understand the structure. In NMR, based on the tris-triazine structure, the peak height of C1 and C2 carbon should be equal; however, the data suggests something very different. In FTIR, author only discuss a single triazine ring.

Response: Thanks for your comments. We agree with you that there is no clear evidence to prove the existence of 3.1 Å holes. As the previous research, the tri-s-triazine can not be polymerized into a perfect crystalline C_3N_4 connected by covalent bonds due to the kinetic problem (Thomas A. et al., *J. Mater. Chem.*, 2008, 18, 4893-4908). Then $-NH_2/NH$ on some incomplete condensed tri-s-triazine rings are tightly connected by hydrogen bonds (Hu Y. et al., *Chem. Mater.* 2017, 29, 5080-5089). As a result, the C_3N_4 shows an amorphous structure. Therefore, as you mentioned in comment 4, it is not necessary to give a certain structure, which may confuse the field. According to your suggestions, we have deleted the corresponding description of triangular pores (3.1 Å) and corrected the description of the structure of C_3N_4 in the revised manuscript.

We agree with you that the peak heights of C1 and C2 carbon should be equal in the ^{13}C NMR spectra based on the tris-triazine structure, theoretically. However, the NMR result shows the peak heights of C1 and C2 carbon are not equal, which is probably because the ^{13}C spectra obtained by the 1H - ^{13}C cross-polarization/MAS (CP-MAS) NMR technique cannot provide information regarding the relative abundance of carbon atoms in different chemical environments according to the signal strength, but rather record the diversity in CP efficiencies. (Wang R. et al., *J. Mater. Chem. A*, 2021, 9, 3985-3994). Actually, the signal intensity ratio varies due to different proximities to

^1H species in the ^1H - ^{13}C CP-MAS mode. The peak of C1 shows higher intensity, which because that C1 is closer to protons of nonpolymerized NH_2 or partially polymerized NH species, where ^{13}C spin polarized NMR signals are greatly enhanced from neighboring ^1H species via a ^1H - ^{13}C dipolar coupling (Hu Y., et al., *Chem. Mater.* 2017, 29, 5080-5089).

As well, according to your suggestion, we have added more discussion about the FTIR results in the revised manuscript. Therein, the characteristic peak located at 810 cm^{-1} is originated from the tri-s triazine breathing mode (Kessler F. et al., *Nat. Rev. Mater.*, 2017, 2, 17030). And the wide bands at $1690\text{-}1150\text{ cm}^{-1}$ and $3680\text{-}2970\text{ cm}^{-1}$ belong to the stretching vibrations of C-N heterocycles and the N-H stretching vibrations of the terminal $-\text{NH}_2/-\text{NH}$ of C_3N_4 , respectively.

According to your comments, we have corrected the description of the structure of C_3N_4 and more discussions about the NMR and FTIR results have been added in the revised manuscript (page 5) and revised supporting information (page 4) as highlighted in yellow.

2. Assuming tris-triazine nanopores are responsible for transport, author carried out DFT simulation of molecules crossing through the nanopore. The DFT calculation predicts a much higher activation energy of CO_2 (0.986 eV) compared to methane (0.694). This is unexpected because CO_2 is a much smaller molecule than methane. Also, based on MD simulation, approximately 120 H_2 and 20 CO_2 molecules cross the pore in 100 ns simulation. This is surprising because the energy barrier for CO_2 is close to 1 eV and in fact with such time scale and high energy barrier, no CO_2 should cross over.

Response: Thanks for your comments. The higher energy barriers for CO_2 (0.986 eV) across the C_3N_4 nanosheets than CH_4 (0.694 eV) may be because not only the size of gas molecules but also the interactions between gas molecules and C_3N_4 were considered in DFT calculations. As shown in **Figure R1**, C_3N_4 nanosheets contain a large number of negatively polarized N atoms (blue reflects negatively polarized sites, red represents the opposite). Nanosheets with such properties show stronger electrostatic interactions with gas molecules and thus influence gas permeability. For example, CO_2 with the deepest red possesses strong interaction with C_3N_4 nanosheets (blue reflects nucleophilic sites), which increases the resistance to CO_2 diffusion. While CH_4 with light red possesses relatively weak interactions with C_3N_4 nanosheets,

resulting in relatively fast CH₄ diffusion. This is consistent with N-functionalized graphene membranes previously reported (Shan M. et. al. *Nanoscale*, 2012, 4, 5477-5482). When the carbon atoms were substituted by nitrogen in the porous graphene membrane, the properties of the nanosheets changed and influenced the gas permeability. Therefore, due to the synergistic influence of size exclusion and the interactions between gas molecules and C₃N₄ nanosheets, the calculated energy barrier of CO₂ is higher than that of CH₄.

Figure R1. Schematic illustrations of gas molecules with electrostatic potentials distribution through the C₃N₄ layer.

The CO₂ molecules can cross the C₃N₄ nanosheets during the MD simulations even though the high energy barriers for CO₂ (0.986 eV) to cross over the nanosheets calculated by DFT simulations, which may be because there is a great difference between the two calculation methods. The potential energy with classical MD simulation only considers the intra and intermolecular potential energies and does not include the internal part of the molecular partition function. However, DFT calculations use different reference states for the energies (separate nucleus, electrons). As a result, in this work, the selectivity results calculated by DFT are higher than that of MD simulations, which also explains why the CO₂ molecules can cross the nanosheets during the MD simulations even though the high energy barriers for CO₂ (0.986 eV) to cross over the nanosheets calculated by DFT simulations.

According to your suggestion, **Figure R1** has been added as **Figure 4a** in the revised manuscript (page 28) and the relevant discussions have been added in the revised manuscript (pages 10, 11, and 12) as highlighted in yellow.

3. MD simulation predicts a H₂/CO₂ selectivity of 6 which is not discussed. This is several orders of magnitude smaller than the selectivity one expects from DFT data (i.e. ratio of $\exp(-0.397 \text{ eV}/kT)$ for H₂ and $\exp(-0.986 \text{ eV}/kT)$ for CO₂). No explanation is provided. I could not find the description of feed pressure for the membrane study. If feed is not pressurized, the data should be reported from at least mildly pressurized feed (transmembrane pressure difference of 2 bar). This is because literature which indicates that the performance of laminar membranes can be sensitive to the feed pressurization.

Response: Thanks for your comments. As shown in **Figure R2**, due to the limitation of MD simulation, only 260 gas molecules (130 for each gas species) are placed in the feed chamber for H₂/CO₂ mixtures. It is found that all H₂ molecules have moved through the C₃N₄ layers at the very beginning, yielding a diffusivity ratio of 130:4 for H₂:CO₂ with the H₂/CO₂ selectivity of 32.5 at 1.0 ns. However, since there is no H₂ molecule on the feed side while CO₂ continues to pass through in the subsequent simulation, hence the separation factor decline with extended time, and the simulations exhibit a H₂/CO₂ selectivity of 6 at 100 ns. This is different from the practical separation process where continuous H₂ is supplied on the feed side. According to your suggestions, we have added more discussions on the MD simulation results in the manuscript.

The theoretical H₂/CO₂ selectivity can be calculated from the DFT data (Here we adopt the latest DFT calculation results considering the periodic boundary conditions) based on the Arrhenius equation defined in equation (1):

$$S_{H_2/CO_2} = \frac{r_{H_2}}{r_{CO_2}} = \frac{A_{H_2} \exp(-E_{H_2}/k_B T)}{A_{CO_2} \exp(-E_{CO_2}/k_B T)} \quad (1)$$

where r is the rate of translocation and A is the ideal gas diffusion prefactor. E_{H₂} and E_{CO₂} are the diffusion energy barrier for H₂ and CO₂, respectively; k_B is the Boltzmann constant and T is the temperature. Assuming that A_{H₂} and A_{CO₂} are within the same order of magnitude, an approximate gas pair selectivity can be calculated, which is shown below.

$$\begin{aligned} S_{H_2/CO_2} &= \frac{A_{H_2} \exp(-E_{H_2}/k_B T)}{A_{CO_2} \exp(-E_{CO_2}/k_B T)} \\ &= \frac{\exp[-0.1322 \times 1.6 \times 10^{-19} / (1.3806505 \times 10^{-23} \times 298)]}{\exp[-0.9687 \times 1.6 \times 10^{-19} / (1.3806505 \times 10^{-23} \times 298)]} \\ &= 1.3 \times 10^{14} \end{aligned}$$

Consequently, the DFT calculations predicted an excellent selectivity of H₂/CO₂

with 1.3×10^{14} at room temperature, which is consistent with the previously reported selectivity of H₂/CO₂ (10^{15}) (Li Y. et al. *RSC Adv.*, 2016, 57, 52377). While the H₂/CO₂ selectivity predicted by MD is only 32.5. The selectivity predicted by these two methods shows several orders of magnitude differences, which may be that DFT calculations use different reference states for the energies (separate nucleus, electrons) while potential energy with classical MD simulation only includes the intra and intermolecular potential energies and not the piece due to internal part of the molecular partition function. There is a great difference between the two calculation methods. Therefore, there is not a strict one-to-one correspondence between the results by DFT calculations and MD simulations, as reported by Wang et al. (Wang J. et al., *App. Surf. Sci.*, 2018, 441, 408-414). Even so, the MD results qualitatively agree well with the DFT results. According to your suggestion, we added the explanation of the differences between DFT results and MD results in the revised manuscript.

Figure R2. The number of gas molecules of H₂/CO₂ mixtures through the C₃N₄ layer in molecular dynamics (MD) simulations as a function of simulation time. Simulation snapshots at 0, 1, and 100 ns for H₂/CO₂ mixtures are given.

In addition, we agree with the reviewer that stability under different feed pressure is important for a membrane. Hence, according to your suggestions, we measured the H₂/CO₂ separation performance of the C₃N₄ membrane under elevated transmembrane pressures by continuously increasing the feed pressure while fixing the pressure in the sweep side at 1 bar (the transmembrane pressure is the difference between the feed pressure and the sweep pressure). As the transmembrane pressure increases, the H₂/CO₂ separation factor gradually decreases (**Figure R3**). This could be attributed to (i) the C₃N₄ membrane structure will be distorted when there is an absolute pressure difference across the membrane due to the nanosheet flexibility, resulting in the increased defective sites in the membrane (Wang P. et al., *Angew. Chem. Int. Ed.*, 2021, 60, 19047-

19052); or (ii) the splendid H₂/CO₂ separation performance of the nanosheet membrane comes from not only the size sieving but also the strong interaction between CO₂ molecules and C₃N₄ (Ashirov T. et al., *Adv. Mater.*, 2022, 34, 2106785). The delayed CO₂ transport caused by the interaction will be greatly reduced with the increment of transmembrane pressures. But the selectivity can be recovered to the initial value basically after releasing the pressure. As well, as the feed pressure increases to 1.4 bar (transmembrane pressure: 0.4 bar), the H₂/CO₂ separation factor has decreased to 1.43, so we do not increase the pressure anymore for the H₂/CO₂ separation tests.

To be more convincing, we further measured the separation performance of H₂ and larger molecules C₃H₈ under elevated transmembrane pressures. As presented in **Figure R4**, the C₃N₄ membrane can be operated at higher feed pressure. When the feed pressure increases from 1 bar to 2 bar (transmembrane pressure increases from 0 bar to 1 bar), both the H₂ and C₃H₈ permeances increase sharply with decreasing H₂/C₃H₈ separation factor from 83 to 17. After pressure release, the H₂/C₃H₈ separation factor can return to the initial value completely, indicating that under the elevated pressure most probably only gate-opening phenomena happen that allow larger molecules to pass through but no cracks appear.

Figure R3. Gas permeance and H₂/CO₂ separation factor of the C₃N₄ membrane as a function of the feed pressure at room temperature.

Figure R4. Gas permeance and H₂/C₃H₈ separation factor of the C₃N₄ membrane as a function of the feed pressure at room temperature.

According to your comments, **Figures R3** and **R4** has been added as **Figures S33** and **S34** in the revised supporting information (pages 35, 36) and the abovementioned relevant discussions have been added to the revised manuscript (pages 9, 10, 11, and 12) as highlighted in yellow.

4. Overall describing the structure of amorphous g-C₃N₄ is extremely challenging. I am of the opinion that it is perhaps not necessary to assume a certain structure because it confuses the field. I therefore recommend (i) either providing more direct proof of structure, or (ii) removing description of structure as made of 3.1 Å pores.

Response: Thanks for your valuable advice. To avoid any confusion for the readers, according to your suggestion, we have removed the description of the C₃N₄ structure with 3.1 Å pores. In fact, researchers have found that some uncondensed tri-s-triazine rings did exist in C₃N₄ due to the kinetic problem and then the terminal -NH/NH₂ in tri-s-triazine rings forms an amorphous structure through hydrogen bonding (Hu Y., et. al, *Chem. Mater.*, 2017, 29, 5080-5089). Accordingly, we have modified the description of the C₃N₄ structure in the revised manuscript (pages 2, 5) as highlighted in yellow.

5. "However, the development of traditional polymer or zeolite membranes is limited by the trade-off relationship between permeability and selectivity (known as Robeson's

upper bounds)”. Please correct this as Robeson upper bound is only for polymers.

Response: Thanks for your comments. We have corrected the sentence to "the development of traditional polymer membranes is limited by the trade-off relationship between permeability and selectivity (known as Robeson’s upper bounds)" in the revised manuscript (page 2) as highlighted in yellow.

6. *The Fourier transform infrared spectroscopy (FTIR) of both types of nanosheets (Supplementary Fig. 4a) showed signals corresponding to the in-plane organization of tri-s-triazine units in C₃N₄”. I do not think that FTIR show in-plane organization but rather FTIR signal from tri-s triazine units.*

Response: Thanks for your comments. We agree with you that the FTIR signal comes from the tri-s-triazine units. Wherein, the characteristic peak located at 810 cm⁻¹ originates from the characteristic tri-s-triazine breathing mode (Kessler F. et al., *Nat. Rev. Mater.*, 2017, 2, 17030). And the wide bands at 1690-1150 cm⁻¹ and 3680-2970 cm⁻¹ belong to the stretching vibrations of C-N heterocycles and the N-H stretching vibrations of the terminal -NH₂/-NH of C₃N₄, respectively. According to your suggestion, we have corrected the corresponding description of FTIR results in the revised manuscript (page 5) as highlighted in yellow.

7. *“the peak at 12.9° corresponding to the (100) periodic repetition of tri-s-triazine verifies the existence of triangular nanopores in the C₃N₄”; This should be explained in terms of structure or schematic. The material prepared is an amorphous material. It is not clear as to what structure relates to the (100) peak. Further, it is not clear as to how this peak relates to pore or pore shape as there is lack of in-plane order in the synthesized material.*

Response: Thanks for your comments. According to your suggestion, the XRD pattern was reanalyzed in terms of the C₃N₄ structure. In the C₃N₄, some tri-s-triazine units are connected by covalent bonds, while some tri-s-triazine units are incompletely polycondensated and then connected by hydrogen bonds through -NH₂/NH at their edges. Consequently, C₃N₄ shows an amorphous structure due to the uncertainty of the connection mode of tri-s-triazine rings in the 2D plane (by covalent bonds or hydrogen bonds). And weak van der Waals force exists between the layers.

As a result, the (100) peak of the XRD pattern results from the lattice planes along c-axis due to the 2D planer disorder as shown in **Figure R5** (Lotsch B. et al., *Chem.*

Eur. J., 2007, 13, 4969-4980), and it cannot prove the existence of triangular holes. Another reflection was observed at $2\theta = 27.46^\circ$, which is attributed to the stacking motif of the very same tri-s-triazine on top of each other. Moreover, as you mentioned in your comment 4, it is not necessary to give a certain structure, which may confuse the field. Therefore, according to your suggestions, we have deleted the corresponding description of triangular pores (3.1 Å) and corrected the description of the structure of C_3N_4 in the revised manuscript.

According to your suggestion, **Figure R5** has been added as **Figure S5** in the revised supporting information (page 6) and we have corrected the analysis results of the XRD pattern in the revised manuscript (page 5) as highlighted in yellow.

Figure R5. Schematic diagram of the hydrogen bond structure in C_3N_4 nanosheets.

Response to Reviewer 3

Zhou et al performed an interesting experimental-theoretical study on H₂ permeability and gas selectivity on carbon nitride nanosheets. The study is comprehensible and well-written, it deserves publication in Nature communication after the authors give full attention to the following comments and suggestions: Major revision.

Response: Thank you for your encouragement and valuable suggestions. We have revised our manuscript carefully according to your suggestions point-by-point and hope these added experiments and explanations could help the readers understand our work more easily.

1. On page 6, the authors mentioned results coming from their DFT calculations. Nevertheless, such information is abruptly given. It is then recommended that the DFT output mentioned in this section should be adequately presented prior to be invoked and compared with experimental data. Such results (line 150-151), as they are given in the manuscript are not understood, and they are challenging to interpret.

Response: Thanks for your comments. We are sorry for the unclear description in Line 150-151. As shown in **Figure R6**, the stacking modes of nanosheets include the aligned AA stacking and unaligned stacking including AB, AC, etc.

Figure R6. (a) Aligned and (b) nonaligned stacking of C₃N₄ nanosheets.

It reported that the aligned AA stacking mode of nanosheets will form the relatively large sieving channels while the unaligned stacking of adjacent layers offset each other will narrow the effective sieving channels of the membrane (Wang P. et. al., *Angew. Chem. Int. Ed.*, 2021, 60, 19047-19052.). In other words, the aligned and unaligned stacking between adjacent layers will affect the separation applications of the membrane. Therefore, it is also important to discuss the stacking modes of nanosheets. The stacking modes of these two kinds of nanosheets were calculated by DFT. As shown in **Figure R7a**, the calculated stacking energy of bilayer bottom-up C₃N₄ shows

that its AA stacking has minimum energy configuration, indicating that the bottom-up C_3N_4 nanosheet tends to the aligned AA stacks in the C_3N_4 film, forming conceivable gas-permeable interlayer pathways. On the contrary, the calculation results of top-down C_3N_4 nanosheets show that they tend to be nonaligned AB stacking when assembled into the membrane (**Figure R7b**). As a consequence, the effective sieving channel of the C_3N_4 membrane is greatly narrowed, which may prevent the transmission of gas.

Figure R7. The DFT calculations about stacking states of two types of C_3N_4 nanosheets prepared through the (a) bottom-up and (b) top-down method in C_3N_4 membranes.

According to your suggestion, **Figures R6** and **R7** have been added to **Figure 2** in the revised manuscript (page 26) and as **Figure S14** in the revised supporting information (page 15), and these additional discussions for the DFT calculation results have been added in the revised manuscript (page 7) as highlighted in yellow. Hopefully, these added results and explanations could help understand the manuscript.

2. *If the Top-down and Bottom-up methods were somehow simulated with the theoretical methodologies, the corresponding molecular arrangements and discussion of the results should be presented in the main text of the manuscript. It is highly recommended that Fig. S14 and S15 be presented in the main text.*

Response: Thanks for your comments. **Figure R8** shows the DFT simulations about the stacking state of top-down and bottom-up nanosheets. The specific operations of the simulation were as follows: Two-layer C_3N_4 systems corresponding to these two kinds of nanosheets were constructed, respectively. Then fix one layer of nanosheets and move the other layer to specific extents. The total energies of these two-layer systems were calculated with different positions. Abscissa was shift steps of the C_3N_4 layer. The initial state of the C_3N_4 system (zero point) was aligned AA stacking mode.

It can be seen that the calculated stacking energy of bilayer bottom-up C_3N_4 shows that its AA stacking has minimum energy configuration, indicating that the bottom-up C_3N_4 nanosheet tends to the aligned AA stacks in the C_3N_4 film, forming conceivable gas-permeable interlayer pathways, as shown in **Figure R9a**. On the contrary, the calculation results of top-down C_3N_4 nanosheets show that they tend to be nonaligned AB stacking when assembled into the membrane. As a consequence, the effective sieving channel of the C_3N_4 membrane is greatly narrowed, which may prevent the transmission of gas (**Figure R9b**).

Figure R8. The DFT calculations about stacking states of two types of C_3N_4 nanosheets prepared through the (a) top-down and (b) bottom-up method in C_3N_4 membranes.

Figure R9. (a) Bottom-up and (b) Top-down nanosheets of C_3N_4 nanosheets.

According to your suggestion, **Figures R8** and **R9** have been added to **Figure 2** in the revised manuscript (page 26) and as **Figure S14** in the revised supporting information (page 15), and these additional discussions have been added in the revised manuscript (page 7) as highlighted in yellow.

3. For completeness of the study, it is suggested that the authors perform MD simulations to test the stability of the C_3N_4 system; i.e., geometry restrictions should be omitted. This could be performed only with a representative system.

Response: Thanks for your comments. According to your suggestion, we have employed the MD simulations to verify the stability of the C_3N_4 materials. **Figure R10** shows the total energy of the C_3N_4 membrane as a function of time. The C_3N_4 system shows very small energy fluctuations during the simulation time scale. This is consistent with the substantially unchanged schematic diagram of C_3N_4 membranes before and after MD simulations (**Figure R11**), reflecting its good structural stability. The experimental results also show the outstanding stability of the C_3N_4 membrane, such as under the conditions of temperature swings, wet atmosphere and long-term operation of more than 200 days.

Figure R10. Total energy for C_3N_4 membranes during the MD simulations.

Figure R11. Schematic diagrams of C_3N_4 layer before and after the MD simulations.

According to your suggestion, **Figures R10** and **R11** have been added as **Figures S31** and **S32** in the revised supporting information (pages 33 and 34) and the relevant discussions and simulations details have been added to the revised manuscript (pages 9, 17) as highlighted in yellow.

4. On page 10, lines 236-238 the authors stated: “CO₂, N₂, and CH₄ molecules (slightly larger than 3.1 Å) can pass through the pores because of the partial overlap of electron clouds of the gas molecules and the atoms around the C₃N₄ triangular pore”. To give insight into this assumption, please map the PDOS of C₃N₄/gas molecules systems to quantify the degree of overlapping.

Response: Thanks for your comments. According to your suggestion, the partial density of states (PDOS) analysis was conducted to elucidate the overlapping degree of C₃N₄ and gas molecular systems through the occupation of the electron energy levels from different gas molecules and the C₃N₄ adsorbed by gas molecules. **Figure R12a** and **R12b** show the orbital occupation and energy distribution of the s, p, d, f orbitals of H₂ adsorbed C₃N₄ and H₂ molecule models, respectively. And the common peak position of the two figures is the overlapping part of the electron orbit for C₃N₄+H₂ molecules systems. Similarly, the overlapped electronic orbitals of C₃N₄+CO₂ (C₃N₄+N₂, or C₃N₄+CH₄) molecular systems are the common peak positions of **Figures 12c** and **12d**, respectively (**Figures 12e** and **12f**, or **Figure. 12g** and **12h**). In addition, the overlapping parts of the electron orbit for C₃N₄ and different gas molecules increase with the increase of gas molecular size.

Figure R12. (a, c, e, f) PDOS of the C_3N_4 adsorbed by gas molecules (H_2 , CO_2 , N_2 , CH_4). (b, d, f, h) PDOS of H_2 , CO_2 , N_2 , CH_4 gas molecules models. The assignment of color: black = s orbit; red = p orbit; blue = d orbit, green = f orbit.

According to your suggestion, **Figure R12** has been added as **Figure S35** in the revised supporting information (page 37), and the abovementioned discussions and simulation details have been added to the revised manuscript (pages 10, 15, and 16) as highlighted in yellow.

5. On page 10, lines 239-240: There seems to be an anomaly with Fig. 4(c), since the E_b of the reaction pathways do not correspond to the E_b of the column graph shown in the inset. If the curves correspond to reaction pathway calculations, please give the Computational details of the method used.

Response: Thanks for your comments. **Fig. 4(c)** shows the interaction energy between C_3N_4 and gas molecules when gas molecules were put at different distances from the nanosheets in the z-direction. Then the energy barriers (E_b , in the inset column graph) for gas molecules passing through the C_3N_4 layer can be calculated by the difference in the interaction energy (E , in **Fig. 4(c)**) at E_{TS} (E_{TS} , defined as the interaction energy at $z=0$) and E_{SS} (E_{SS} , defined as the maximum attractive interaction energy when $z \neq 0$).

The computational details of the method used to calculate E_b are as follows. Firstly, the potential energy surfaces for different gases passing through the C_3N_4 monolayer were explored by calculating the interactions energy with the C_3N_4 membrane, where the gas molecule was put at different distances from the sheet, and the mass center of

the molecule was fixed in the z-direction (perpendicular to the nanoporous two-dimensional nanosheet).

$$\text{Interaction energy} = E_{C_3N_4\text{-gas}} - (E_{C_3N_4} + E_{gas})$$

where $E_{C_3N_4\text{-gas}}$ is the total energy of the C_3N_4 -gas configuration, $E_{C_3N_4}$ and E_{gas} are the single point energy of C_3N_4 and the gas molecules, respectively.

Next, the E_b (energy barrier) is defined as the difference between the interaction energies at the transition state (TS, $z=0$) and the stable state (SS, the stable adsorption state when the attractive interaction is maximum and $z \neq 0$) of gas molecule on C_3N_4 .

$$E_b = E_{TS} - E_{SS}$$

where E_{TS} and E_{SS} , respectively, stand for the TS energy and SS energy when a gas molecule permeates through the C_3N_4 nanosheet, as shown in **Figure R13**.

The specific data of transition state energy E_{TS} (eV), stable state energy E_{SS} (eV) and corresponding height (\AA), and the calculated diffusion energy barriers of gas molecules passing through the C_3N_4 layer are summarized in **Table R1**. The calculated E_b values for gas molecules passing through the nanosheets are 0.132, 0.969, 0.782, and 0.791 eV for H_2 , CO_2 , N_2 , and CH_4 , respectively.

Figure R13. Potential energy surfaces for H_2 , CO_2 , N_2 , CH_4 on the C_3N_4 layer. The red dotted line indicates the TS and SS of CH_4 molecules

Table R1. The transition state energy E_{TS} (eV), stable state energy E_{SS} (eV) and energy barriers E_b (eV) of gas molecules on the C_3N_4 lattice.

	Transition state		Stable state		E_b (eV)
	Height (\AA)	E_{TS} (eV)	Height (\AA)	E_{SS} (eV)	
H_2	0	0.0346	1	-0.0976	0.1322

CO ₂	0	0.7691	3	-0.1996	0.9687
N ₂	0	0.5485	2	-0.2339	0.7824
CH ₄	0	0.5458	1	-0.2447	0.7905

According to your suggestion, we have emphasized the computational details of the method for calculation of E_b in the revised manuscript (page 16) and **Table R1** has been added as **Table S6** in the supporting information (page 44) as highlighted in yellow.

6. On page 10, lines 243-244, the authors stated: “Obviously, their transport behaviors are not only affected by molecular size but also by interactions between the gas molecules and C₃N₄ nanosheets”. This is not necessarily obvious; the authors are requested to verify how the electronic states are overlapped among the C₃N₄ and the different gas molecules. Additionally, the authors should comment Figs. S31 and S32 and deepen into the issue. It is not enough to only indicate where the figure is.

Response: Thanks for your comments. According to your suggestion, we have conducted the partial density of states (PDOS) analysis to explain the overlap of electron clouds between C₃N₄ and different gas molecules. In addition, we have added more discussions about the results of **Figures S31** and **S32** to clarify the interactions between gas molecules and C₃N₄ nanosheets.

Firstly, PDOS results of gas molecules and the C₃N₄ adsorbed by gas molecules are shown in **Figure R14**. **Figure R14a** and **R14b** show the orbital occupation and energy distribution of the s, p, d, f orbitals of H₂ adsorbed C₃N₄ and H₂ molecule models, respectively. And the common peak position of the two figures is the overlapping part of the electron orbit for C₃N₄+H₂ systems. Similarly, the overlapped electronic orbitals of C₃N₄+CO₂ (C₃N₄+N₂, or C₃N₄+CH₄) molecular systems are the common peak positions of **Figure R14c** and **R14d**, respectively (**Figure R14e** and **R14f**, or **Figure R14g** and **R14h**). Consequently, H₂, CO₂, N₂ and CH₄ molecules can pass through the C₃N₄ nanosheets after the partial overlap of electron clouds of the gas molecules and the atoms around the C₃N₄ nanosheets.

Figure R14. (a, c, e, f) PDOS of the C_3N_4 adsorbed by gas molecules (H_2 , CO_2 , N_2 , CH_4). (b, d, f, h) PDOS of H_2 , CO_2 , N_2 , CH_4 gas molecules models. The assignment of color: black = s orbit; red = p orbit; blue = d orbit, green = f orbit.

In addition, we have added more discussions on **Figures R15 and R16 (Figures S31 and S32)** in the revised manuscript, as shown below. From the schematic illustration of the electrostatic potential distribution of gas and C_3N_4 models shown in **Figure R15**, it can be seen that C_3N_4 nanosheets contain a large number of negatively polarized N atoms (blue reflects negatively polarized sites, red represents the opposite). Nanosheets with such properties show stronger electrostatic forces with gas molecules and thus influencing the gas permeability. For example, CO_2 with the deepest red possesses strong interaction with C_3N_4 nanosheets (blue reflects nucleophilic sites), which increases the resistance to CO_2 diffusion and results in a high separation factor

of H₂/CO₂. While CH₄ with light red possesses relatively weak interactions with C₃N₄ nanosheets, resulting in relatively fast CH₄ diffusion. This is consistent with N-functionalized graphene membranes (Shan M. et. al. *Nanoscale*, 2012, 4, 5477-5482). When the carbon atoms were substituted by nitrogen in the porous graphene membrane, the properties of the membrane changed and influenced the gas permeability. The adsorption isotherms of H₂, CO₂, N₂, CH₄ on the C₃N₄ membranes at room temperature also indicate that the C₃N₄ membranes tend to preferentially adsorb CO₂ compared to other gases, such as H₂, N₂, CH₄ (**Figure R16**), which delays CO₂ transport.

Figure R15. Schematic illustration of gas molecules with electrostatic potentials distribution through the C₃N₄ nanosheets.

Figure R16. Adsorption isotherms of H₂, CO₂, N₂ and CH₄, on C₃N₄ nanosheets at room temperature.

According to your suggestion, **Figure R14** has been added as **Figure S35** in the revised supporting information (page 37), and the abovementioned discussions about

the results of **Figures S31 and S32** have been added to the revised manuscript (pages 10, 11) as highlighted in yellow.

7. Regarding the DFT computational details (page 14): Although the methodology is correctly applied, this reviewer suggests that the authors consider the border effects; i.e., the truncation of C_3N_4 and the addition of H-atoms to complete valences may play a non-beneficial role in the computations. The authors are urged to perform a calculation in which periodic boundary conditions are included, where the border effects are excluded.

Response: Thanks for your helpful advice. According to your suggestion, to avoid the border effects, the interaction energies between gas molecules and the C_3N_4 monolayer were recomputed by DFT with periodic boundary conditions (Lin L. et al., *J. Chem. Theory Comput.* 2014, 10, 1477-1488). After removing the possible effects of C_3N_4 truncation and the added H-atoms in DFT calculations, the corrected potential energy surfaces and Eb are shown in **Figure R17**. The calculated Eb values for gas molecules passing through the nanosheets are 0.132, 0.969, 0.782, and 0.791 eV for H_2 , CO_2 , N_2 , and CH_4 , respectively (**Table R2**). The simulated gas permeation trend is approximately consistent with the experimental results, where the permeance order is $H_2 > N_2 > CO_2$. However, the permeation of CH_4 was slightly higher than that of N_2 in the experiment, which is different from the simulation results. This may be because the deviation in the experimental measurement or CH_4 as a spherical molecule is easier to pass through.

Figure R17. Potential energy surfaces for H_2 , CO_2 , N_2 , CH_4 on the C_3N_4 layer. Inset is the energy barrier for gas molecules across the nanosheets.

Table R2. The transition state energy E_{TS} (eV), stable state energy E_{SS} (eV) and energy barriers E_b (eV) of gas molecules on the C_3N_4 lattice.

	Transition state		Stable state		E_b (eV)
	Height (Å)	E_{TS} (eV)	Height (Å)	E_{SS} (eV)	
H ₂	0	0.0346	1	-0.0976	0.1322
CO ₂	0	0.7691	3	-0.1996	0.9687
N ₂	0	0.5485	2	-0.2339	0.7824
CH ₄	0	0.5458	1	-0.2447	0.7905

According to your suggestion, **Table R2** has been added as **Table S6** in the revised supporting information (page 44). **Figure R17** has been used as **Figure 4b** in the revised manuscript (page 28) and more discussions were added in the revised manuscript (page 11) as highlighted in yellow.

8. On page 15, line 366: Please give more specifications of the force field. Is it universal? Which module in Materials Studio was implemented to perform the MD simulations?

Response: Thanks for your comments. The Dreiding force field was adopted during the MD simulation and it is universal. The Forcite module in Materials Studio was implemented to perform the MD simulation. According to your suggestion, we have emphasized that the Dreiding force field was adopted during the MD simulation and MD simulation was performed by the Forcite module in Materials Studio in the revised manuscript (page 17) as highlighted in yellow.

Response to Reviewer 4

In this manuscripts, the authors reported on the C₃N₄-based membrane for selective H₂ filtration against several other gas molecules. The work is overall of high importance, using C₃N₄ as the active layer in the filtering membrane which shows high performance in gas separation and purification. However, I have the below concerns before considering the manuscript to be published:

Response: Thank you for your encouragement and kind comments. We have revised our manuscript prudently according to your suggestions point-by-point and we hope these added experiments and explanations could help the readers understand our work more easily.

1) The C₃N₄ is claimed to be "defect-free", which is quite ambiguous and unfair since C₃N₄ has certain intrinsic structural defects arising from high temperature synthesis. No strong evidence is given to show the material is "perfect"

Response: Thanks for your comments. According to your suggestion, the inappropriate descriptions of “defect-free” have been corrected to “high quality” in the revised manuscript and supporting information as highlighted in yellow.

2) Any adsorption of H₂ or other gases by the membrane?

Response: Thanks for your suggestions. The adsorption isotherms of H₂, CO₂, N₂, CH₄ on the C₃N₄ membranes at room temperature were analyzed. As shown in **Figure R18**, the C₃N₄ membranes tend to preferentially adsorb CO₂ compared to other gases, such as H₂, N₂, CH₄. It can be explained that the -NH or -NH₂ groups on the C₃N₄ form strong interactions with CO₂ molecules, which delays CO₂ transport rates through C₃N₄ membranes and thus results in the increased H₂/CO₂ separation factor. This is consistent with previous reports (Hou J. et al., *Chem. Eng. Sci.*, 2018, 182: 180-188).

Figure S18. Adsorption isotherms of H₂, CO₂, N₂ and CH₄, on C₃N₄ nanosheets at room temperature.

According to your comments, **Figure R18** has been added as **Figure S36** in the revised supporting information (page 38) and more discussions have been added in the revised manuscript (page 11) as highlighted in yellow.

3) *How about the mechanical strength of the membranes?*

Response: Thanks for your suggestion. According to your suggestion, the mechanical properties of the bare PES substrates and C₃N₄ membranes with different thicknesses supported on the PES substrates were investigated. As shown in **Figure R19**, compared with that of the bare PES substrate, the PES-supported C₃N₄ membranes exhibit much higher tensile strength and lower percentage elongation. With increasing membrane thickness from 0.15 μm to 1 μm, the Young's modulus increases continuously (**Table R3**), confirming the enhanced mechanical strength of the C₃N₄ membranes. Therefore, the C₃N₄ membranes exhibit excellent mechanical properties.

Figure R19. Stress-strain curves of the PES-supported C₃N₄ membranes with different thicknesses from 0.15 μm to 1 μm, and the bare PES substrate.

Table R3. Mechanical properties of the PES-supported C₃N₄ membranes and the bare PES substrate.

Membrane	Thickness (μm)	Strain (%)	Tensile stress (MPa)	Young's modulus (MPa)
PES	0	60.86	6.18	92.29
C ₃ N ₄ /PES	0.15	43.06	6.64	99.9
C ₃ N ₄ /PES	0.3	49.28	7.11	106.09
C ₃ N ₄ /PES	0.7	52.98	6.94	131.83
C ₃ N ₄ /PES	1	46.5	7.43	158.95

According to your comments, **Figure R19** and **Table R3** have been added as **Figure S18** and **Table S2** in the revised supporting information (pages 19, 40), and more discussions about the mechanical property of the C₃N₄ membrane have been added in the revised manuscript (page 8) as highlighted in yellow.

4) *Can the membranes work at elevated temperatures?*

Response: Thanks for your suggestions. Our C₃N₄ membranes can work at elevated temperatures and exhibit excellent stability. The C₃N₄ membrane was exposed to an equimolar H₂/CO₂ mixture for three temperature cycles operation. The results show the C₃N₄ membrane could be operated up to 150°C (**Figure R20**). As the test temperature increased from 25 to 150 °C, the permeances of both H₂ and CO₂ increased, but the rate

of increasing CO₂ permeance is fast, resulting in a decrease in the H₂/CO₂ separation factor, which is due to the higher activated diffusion energy of CO₂ compared to H₂. However, the H₂/CO₂ mixed separation factor of the C₃N₄ membrane can almost recover to its initial level after the test temperature returned to room temperature. In addition, long-term operational stability under humid conditions of the C₃N₄ membrane was evaluated (**Figure R21**). The membrane can perform well even under wet gas mixture conditions (water activity of 0.353) at 120 °C for over 100 h, indicating its good hydrothermal stability.

Figure R20. Three temperature cycles operation for a 1- μm -thick bottom-up C₃N₄ membrane with equimolar H₂/CO₂ mixture. The permeances of the bottom-up C₃N₄ membrane in the temperature-swing fluctuate slightly, however, the separation factor all return to the original level.

Figure R21. Long-term stability for H₂/CO₂ separation under dry and wet gas mixture (water activity of 0.353, marked by blue areas) at 120 °C and 1 bar.

According to your comments, we have emphasized the excellent thermal stability of the C₃N₄ membranes in the revised manuscript (page 9), and **Figures R20 and R21** have been used as **Figures S28 and S30** in the revised supporting information (pages 30, 32) as highlighted in yellow.

The end

Reviewer #2 (Remarks to the Author):

The authors followed up on the comments and provided detailed response and justification. Most of the response are satisfactory except that on pressurization. The gas separation application in this manuscript focuses on H₂/CO₂ separation. This separation takes place at high pressure (> 10 bar). I had asked author to carry out separation at least at the mild pressure (around 2 bar) to understand the role of defects in the membranes.

Authors carried out this test and found that the separation performance is completely lost when the feed pressure increases from 1.0 to 1.4 bar. This indicates that such membranes will likely not see application in membrane-based separation.

My central point is that the manuscript is quite interesting from the fundamental synthesis and film fabrication point of view. However, the gas separation performance of the resulting membranes is not good. Comparison with literature (including with polymeric membranes) are done in Robeson plot (Fig. 3d) where data at 0 transmembrane pressure difference is used.

On the fundamental front, author describe the loss of separation performance to nanosheet flexibility, and increased adsorption of CO₂ at higher pressure (1.4 bar vs 1.0 bar). I do not agree with this justification. Feed pressure of 1.4 bar is too small to expand the pore relative to when feed pressure is 1.0 bar. In any case pore flexibility as a function of pressure (1 bar vs 1.4 bar) can be cross-checked by MD simulation. Also with extremely large adsorption energy of CO₂ predicted by DFT, CO₂ should already saturate most sites at 1 bar. As a result, adsorbed concentration of CO₂ at 1.4 bar should not increase compared to the case at 1.0 bar. This can be also verified by MD simulation. In my opinion, the most likely reason behind loss in selectivity should be parallel nonselective transport pathways where viscous diffusion is prevalent, and which dominates the membrane performance when transmembrane pressure difference is greater than 0.

Given the other aspects of the manuscript is highly interesting, I would recommend publication of the manuscript only if the author include Fig. R3 and R4 in the main text (not in SI), describing and explaining the loss in selectivity after pressurization for the benefit of the field.

Reviewer #3 (Remarks to the Author):

Dear editor

The authors have addressed all issues raised by this reviewer. The publication of the manuscript is recommended.

Reviewer #4 (Remarks to the Author):

The authors have made significant improvement to the manuscript by conducting many experiments and calculations as suggested by the reviewers. I have carefully checked and my feeling is that it can be published in the present form. However, before that, I would strongly suggest the authors add some highly relevant papers regarding carbon nitrides membranes. These papers may not work on gas separation but rather on water treatment and other applications, but their priorities should be fully respected. For example, *Angew. Chem. Int. Ed.*, 130 (32) (2018), pp. 10280-10283; *J. Hazard. Mater.*, 365 (2019), pp. 606-614; *J. Membr. Sci.*, 490 (2015), pp. 72-83. The authors could refer to this review article for more information: <https://doi.org/10.1016/j.watres.2021.117207>

Response to the Reviewers' Comments

Many thanks to the reviewers for their valuable comments and suggestions. The followings are the point-by-point answers to the concerns:

Response to Reviewer 2

The authors followed up on the comments and provided detailed response and justification. Most of the response are satisfactory except that on pressurization. The gas separation application in this manuscript focuses on H₂/CO₂ separation. This separation takes place at high pressure (> 10 bar). I had asked author to carry out separation at least at the mild pressure (around 2 bar) to understand the role of defects in the membranes.

Authors carried out this test and found that the separation performance is completely lost when the feed pressure increases from 1.0 to 1.4 bar. This indicates that such membranes will likely not see application in membrane-based separation.

My central point is that the manuscript is quite interesting from the fundamental synthesis and film fabrication point of view. However, the gas separation performance of the resulting membranes is not good. Comparison with literature (including with polymeric membranes) are done in Robeson plot (Fig. 3d) where data at 0 transmembrane pressure difference is used.

On the fundamental front, author describe the loss of separation performance to nanosheet flexibility, and increased adsorption of CO₂ at higher pressure (1.4 bar vs 1.0 bar). I do not agree with this justification. Feed pressure of 1.4 bar is too small to expand the pore relative to when feed pressure is 1.0 bar. In any case pore flexibility

as a function of pressure (1 bar vs 1.4 bar) can be cross-checked by MD simulation. Also with extremely large adsorption energy of CO₂ predicted by DFT, CO₂ should already saturate most sites at 1 bar. As a result, adsorbed concentration of CO₂ at 1.4 bar should not increase compared to the case at 1.0 bar. This can be also verified by MD simulation. In my opinion, the most likely reason behind loss in selectivity should be parallel nonselective transport pathways where viscous diffusion is prevalent, and which dominates the membrane performance when transmembrane pressure difference is greater than 0.

Given the other aspects of the manuscript is highly interesting, I would recommend publication of the manuscript only if the author include Fig. R3 and R4 in the main text (not in SI), describing and explaining the loss in selectivity after pressurization for the benefit of the field.

Response: Thank you very much for your encouragement and valuable suggestions. We agree with your analysis of the reason behind loss in selectivity. It was also found that some small imperfections in the stacking of 2D nanosheets may form the nonselective transport pathways during the two dimensional membrane fabrication, such as the GO membrane (Park J. et al., *ACS Omega*, 2022, 7, 16568-16575) and the MoS₂ membrane (Lu X. et al., *Environ. Sci. Technol.*, 2020, 54, 9640-9651). Therefore, as you mentioned, the H₂ and CO₂ permeances increase rapidly with the increased feed pressure, which indicates the presence of nonselective transport pathways in the C₃N₄ membrane (Xing W. et al., *Chem. Eng. J.*, 2022, 442, 136336). Therefore, the possible reason behind the decrease in selectivity at high pressure might be the existence of the parallel nonselective transport pathways where viscous diffusion is prevalent and which dominate the membrane performance when the transmembrane pressure difference is greater than zero (Zhang Y. et al., *J. Membr. Sci.* 2019, 572, 567-579; Zhou R. et al., *Sep. Purif. Technol.*, 2019, 209, 946-954). As a result, the H₂/CO₂ selectivity decreases as feed pressure increases.

According to your comments, we have added the abovementioned discussions of the reason behind the decrease of selectivity in the revised manuscript (page 10). The results of H₂/CO₂ and H₂/C₃H₈ pressure experiments shown in **Figure R1** have been

added to **Figure 3** in the main text of the revised manuscript (page 28), and related discussions have been added to describe and explain the decrease in selectivity after pressurization in the revised manuscript (page 10) as highlighted in yellow.

Figure R1. Gas separation performance through the C_3N_4 membranes. **a)** Single gas permeation of the 1- μm thick C_3N_4 membranes at room temperature and 1 bar. The inset shows the gas selectivity for H_2 over other gases through the C_3N_4 membranes. **b)** Permeance and separation factors of the 1- μm thick C_3N_4 membranes in the equimolar mixed-gas permeation at room temperature and 1 bar (inset shows various gas molecules and kinetic diameters). **c)** Comparison of H_2/CO_2 separation performance of C_3N_4 membranes with state-of-the-art membranes at room temperature. **d)** Long-term stability of the C_3N_4 membrane for equimolar H_2/CO_2 mixture at room temperature and 1 bar. **e)** Gas permeances and H_2/C_3H_8 separation factor of the C_3N_4 membrane as a function of the feed pressure at room temperature. **f)** Gas permeances and H_2/CO_2 separation factor of the C_3N_4 membrane as a function of the feed pressure at room temperature.

Response to Reviewer 3

The authors have addressed all issues raised by this reviewer. The publication of the manuscript is recommended.

Response: Thank you very much for your positive evaluation of our manuscript.

Response to Reviewer 4

*The authors have made significant improvement to the manuscript by conducting many experiments and calculations as suggested by the reviewers. I have carefully checked and my feeling is that it can be published in the present form. However, before that, I would strongly suggest the authors add some highly relevant papers regarding carbon nitrides membranes. These papers may not work on gas separation but rather on water treatment and other applications, but their priorities should be fully respected. For example, *Angew. Chem. Int. Ed.*, 130 (32) (2018), pp. 10280-10283; *J. Hazard. Mater.*, 365 (2019), pp. 606-614; *J. Membr. Sci.*, 490 (2015), pp. 72-83. The authors could refer to this review article for more information: <https://doi.org/10.1016/j.watres.2021.117207>.*

Response: Thanks for the kind comments. According to your comments, we have added some highly relevant papers regarding carbon nitride membranes in water treatment and other applications (Cui Y. et al., *Water Res.*, 2021, 200, 117207; Antonietti M. et. al., *Angew. Chem. Int. Ed.*, 2018, 57, 10123; Zhang Y. et. al., *J. Hazard. Mater.*, 2019, 365, 606; Pan F. et. al., *J. Membr. Sci.*, 2015, 490, 72) in the revised manuscript as highlighted in yellow.